

# Assessing Nonlinear Responses of Low-Level Warm Clouds Under the Impacts of Absorbing Aerosols Using the Cloud-Aerosol Mixing Ratio

Po-Hsun Lin[1], Sheng-Hsiang Wang[1, 2], Otto Klemm[1], Neng-Huei Lin[1, 2]

[1]Department of Atmospheric Sciences, National Central University, Taoyuan, Taiwan
[2]Center for Environmental Monitoring and Technology, National Central University, Taoyuan, Taiwan

*Correspondence to*: Sheng-Hsiang Wang (carlo@g.ncu.edu.tw)

**Abstract.** Air masses carrying absorbing aerosols play a dual role by altering ambient supersaturation and acting as cloud condensation nuclei (CCN), thus contributing to both the semi-direct and indirect aerosol effects. However, in real cloud development processes, aerosol influences can simultaneously act to suppress cloud formation or enhance droplet production,
resulting in microphysical characteristics that remain difficult to describe. In the study, an aerosol and cloud microphysics experiment, focusing on warm cloud events strongly coupled with biomass-burning aerosols transported from the Southeast Asia Peninsula, was conducted in the spring of 2024 at the Lulin Atmospheric Background Station (LABS, 2862m a.s.l) in Taiwan. One-minute resolution of in-situ data was used in the data analysis, allowing exploration of the microphysical responses under varying mixing states of absorbing aerosol concentration and cloud liquid water content. This study applies
the cloud-aerosol mixing ratio (e.g., mass concentration mixing ratio, MCr) in conjunction with the ACI index to describe the behavior of aerosol-cloud interactions better. Results identify two distinct responses: (1) under high MCr conditions (MCr > 4.5), clouds exhibit features of the Twomey effect (ACI ≈ 0.08); (2) under low MCr conditions (MCr = 3-4.5), high aerosol loading coincides with elevated temperatures, lower relative humidity, and a reduction in small droplets (<10 μm), consistent with a semi-direct effect. This study offers a comprehensive explanation of how aerosols affect cloud systems in East Asia. It
also underscores the crucial role of cloud-aerosol mixing ratios in characterizing nonlinear cloud microphysical responses.

## 1 Introduction

Anthropogenic aerosols continue to impact the cloud amount and distribution, which in turn affects the radiative balance and contributes to climate change (IPCC, 2023). The mechanisms through which aerosols influence cloud properties remain highly complex. Since the release of AR5 (IPCC, 2014), the understanding of aerosol effects has been redefined and broadly
categorized into aerosol-radiation interactions (ARI) and aerosol-cloud interactions (ACI). The adjustment processes associated with ARI, as the semi-direct effect (Spracklen et al., 2011; Bond et al., 2013; Boucher et al., 2013; Kacarab et al., 2020), can lead to reductions in cloud amount. This response, driven by aerosol-induced environmental changes, constitutes one of the major pathways through which aerosols alter the radiation budget. On the other hand, scientists have studied the impact of ACI over the past 30 years, showing that low-level warm clouds, which generally exhibit higher horizontal
extensions, are more sensitive to variations in anthropogenic aerosol concentrations (Warren et al., 1988; Wood, 2012; Fan et al., 2016). When aerosol loading increases under constant liquid water path (LWP) conditions, a greater number of cloud





droplets with smaller droplet sizes will form, leading to an increase in cloud optical thickness, which is called the Twomey effect (Twomey, 1974, 1977; Feingold et al., 2001). Additionally, smaller droplet sizes reduce the collision efficiency, thereby lowering the likelihood of rain formation and extending the cloud's lifetime, which, on average, increases cloud albedo

(Albrecht, 1989; Kaufman et al., 2005). Satellite Observations (Christensen et al., 2016; Toll et al., 2017; Chen et al., 2021) and ground-based field campaigns (Zhao et al., 2018; Haywood et al., 2021) have confirmed these theories, and there is a considerable consensus ACI.

However, even though most studies agree that increased aerosol concentrations generally lead to more numerous and smaller cloud droplets, there remains disagreement regarding whether such microphysical changes result in an overall increase

in cloud amount, as the reduction in droplet size may instead enhance evaporation (Chen et al., 2012; Fan et al., 2016; Toll et al., 2019). It is worth noting that such reductions in cloud cover in ARI adjustment may contrast with the tendencies typically attributed to ACI. Irrespective of whether smaller droplets persist in the ambient environment, the decrease in supersaturation during ARI adjustment processes undoubtedly promotes cloud dissipation. These contrasting effects highlight the substantial uncertainty and complexity inherent in aerosol effects on clouds.

Beyond the contrasting effects of ACI and ARI, a deeper layer of complexity arises from the behavior of aerosol-cloud responses, which are often nonlinear and highly sensitive to aerosol properties such as hygroscopicity and absorptivity. While it is widely acknowledged that increased aerosol concentrations generally lead to more numerous and smaller cloud droplets, this relationship is not necessarily linear. Recent studies suggest that beyond a certain threshold of aerosol loading, further increases in aerosol concentrations may have a diminished or even negligible effect on cloud droplet size and number (Feingold

et al., 2001; Saponaro et al., 2017; Chen et al., 2021). Some cloud seeding studies have even indicated that large amounts of hygroscopic aerosols are ineffective at increasing cloud amount or droplet size under limited liquid water content (Silverman and Sukarnjanaset, 2000; Lin et al., 2023). This nonlinearity adds a layer of complexity to interpreting aerosol effects on clouds, particularly when attempting to quantify aerosol-induced changes in cloud water content. The situation becomes even more intricate with the presence of absorbing aerosols such as black carbon. These aerosols can modify the thermal structure of the

atmosphere through radiative heating, which in turn influences cloud formation and dissipation pathways (Bond et al., 2013; Bender et al., 2019). Although the activation of aerosols into cloud condensation nuclei (CCN) is governed by their size and hygroscopicity (Wang et al., 2010; Klemm and Lin, 2016), the presence of absorbing aerosols may suppress supersaturation levels and introduce thermodynamic feedbacks that counteract the microphysical pathways traditionally associated with ACI, which is, as indicated before, the main reason responsible for ARI adjustment. These insights emphasize that aerosol

concentration thresholds and the presence of absorbing species must be carefully accounted for in any attempt to characterize aerosol effects on clouds, as they fundamentally alter the pathways and magnitude of cloud responses to aerosol perturbations.

To quantify the impact of aerosols on cloud properties, previous studies have utilized aerosol-cloud interaction (ACI) index as a measure (Kaufman and Fraser, 1997; Feingold et al., 2001; McComiskey et al., 2009; Lihavainen et al., 2010; Zheng



et al., 2020; Chen et al., 2021). However, many traditional applications of the ACI index rely on the assumption of constant liquid water content (LWC), under which retrieved values based on cloud effective radius ($r_e$) or cloud optical thickness (COT) become comparable. Unless cloud droplet number concentration ($N_d$) is used—a metric more directly tied to aerosol activation efficiency—the applicability of ACI indices remains limited to specific liquid water regimes (McComiskey et al., 2009). Furthermore, as discussed above, variations in aerosol composition, number concentration, hygroscopicity, and optical properties significantly complicate the retrieval and interpretation of ACI signals. These factors contribute to the persistent difficulty in robustly quantifying aerosol effects on clouds. Consequently, observational ACI estimates frequently demonstrate significant spatial and temporal constraints, and may not generalise effectively across varied cloud regimes or environmental conditions (McComiskey et al., 2009; Lihavainen et al., 2010). To present, the accurate quantification of the impact of aerosol on clouds remains one of the most pressing and unresolved challenges in climate change research (Bender et al., 2019; Bellouin et al., 2020).

This study posits that constraining aerosol properties is essential for robust analyses of the relationship between aerosol concentration and cloud water content. In recent years, a series of studies have been conducted over the southeastern Atlantic Ocean to investigate the emission, transport, and cloud-related impacts of biomass-burning aerosols originating from continental sources (Kaufman et al., 2005; Bond et al., 2013; Herbert et al., 2020; Taylor et al., 2020; Haywood et al., 2021). Biomass-burning aerosols, which contain substantial amounts of light-absorbing species such black carbon (BC), complicate aerosol effects. Observational studies have also shown that the water content in the atmosphere plays a critical role in determining whether such aerosols ultimately enhance or suppress cloud formation (Zhang and Zuidema, 2019, 2021). Taiwan is situated downwind of biomass burning emissions from the Southeast Asia Peninsula during spring (Wang et al., 2007; Lee et al., 2011; Lin et al., 2013; Yen et al., 2013; Nguyen et al., 2020), accompanied by cold frontal systems (Huang et al., 2020). This geographic advantage allows for observing cloud system development and impacts influenced by aerosols. Lulin Atmospheric Background Station, located in Taiwan's mountainous region, provides a valuable opportunity to capture and isolate frontal systems influenced by biomass-burning aerosols (Tsay et al., 2016; Pani et al., 2022).

Therefore, this study used in-situ observational instruments, including the Cloud Droplet Probe and 11-D optical particle counter, to investigate a frontal event occurring from March 1 to 12, 2024. Cloud droplet number concentration ($N_d$) is mainly employed to calculate the ACI index. Unlike traditional metrics such as effective radius ($R_e$) or cloud optical thickness (COT), which rely on the assumption of constant liquid water content (LWC), $N_d$ offers a more direct representation of aerosol activation efficiency without requiring constraints on cloud water content (McComiskey et al., 2009). Furthermore, this study proposes the use of the cloud water-aerosol mixing ratio to systematically examine the transitional processes between Twomey effect-dominated ACI responses and ARI adjustments (semi-direct effect) under the influence of absorbing aerosols. This approach moves beyond conventional constraints, offering a unified perspective to disentangle the intertwined microphysical and thermodynamic pathways governing aerosol effects on clouds. The findings offer a crucial advancement in understanding



how different cloud water-aerosol mixing states regulate cloud microphysical processes, thereby contributing to a more physically based and comprehensive assessment of aerosol-cloud-climate linkages. Sect. 2.1 and 2.2 describe the observation site and the instruments used, while Sect. 2.3 to 2.6 present the analytical methods. Sect. 3 presents the relationship between the cloud water-aerosol mixing ratio and the ACI index, and Sect. 3.5 further integrates changes in meteorological conditions as well as the size distributions of cloud droplets and aerosols. Sect. 4 provides the conclusions.

## 2 Data and Method

### 2.1 Observation Site

Lulin atmospheric background station (LABS), which is located at central Taiwan (23.47 °N, 120.87 °E, 2862 m above sea level, a.s.l.), was selected as the site for the field campaign. LABS is an atmospheric background indicator station in East Asia that benefits from its geographical location and altitude (Lee et al., 2011; Cheng et al., 2013; Hsiao et al., 2017; Pani et al., 2022). Most studies have shown that biomass burning aerosol transported from the South East Asia Peninsula can be detected during spring season (Chuang et al., 2014; Hsiao et al., 2017; Huang et al., 2020; Nguyen et al., 2020). LABS not only includes measurements of aerosol mass concentration (i.e., $PM_{2.5}$, $PM_{10}$), optical properties, and carbon mass concentration, but also contains basic atmospheric measurements (i.e., temperature, relative humidity, pressure, wind direction, wind speed), which help to understand the weather conditions.

### 2.2 Data Set

From March 1 to 12, 2024, a field campaign was carried out at LABS to collect extensive data on biomass-burning aerosols and cloud-related meteorological conditions. An Aerosol-Cloud Microphysics Monitoring System, including Cloud Droplet Probe (CDP) and 11-D Optical Particle Counter, was deployed to provide high-resolution measurements of aerosol and cloud size distribution and number concentration (Fig. 1 (a)). Combined with the basic atmospheric measurements and aerosol system at LABS, a comprehensive list of all observed parameters is provided in Table 1.

CDP is a forward-scattering spectrometer manufactured by Droplet Measurement Technologies (DMT). It is designed to be lightweight and mounted on airborne platforms, enabling measurements of cloud droplet number concentration and size distribution. Therefore, a ducted facility is necessary to ensure the required airflow velocity when CDP is positioned on the ground (Fig. 1 (b)). CDP uses a 658 nm single-mode diode laser to detect forward-scattered light at angles between 4° and 12° from cloud droplets passing through the laser beam's effective sampling area of 0.267 mm². Based on Mie scattering theory, the instrument retrieves the size of cloud droplets with diameters ranging from 2 μm to 50 μm. According to Lance (2012), CDP exhibits an overall uncertainty of 20 to 30 % when cloud number concentration reaches 500 cm$^{-3}$, with a tendency to underestimate the number of cloud droplets and overestimate the size cloud droplets. During the observation period, CDP instrument required regular cleaning of its optical windows, with more frequency wiping necessary during rainfall. The raw





data are recorded at a temporal resolution of 1 Hz and averaged to 1-minute intervals for comparison with aerosol size spectrometers and meteorological data from LABS. All measurements were also corrected for variations in airflow velocity within the wind duct, which can be affected by ambient wind.

The 11-D optical particle counter is a portable instrument developed and manufactured by Grimm Aerosol Technik, Germany. Figure 1 (c) shows the instruments' appearance during the field campaign. 11-D utilizes a diode laser with a narrow wavelength band and automatically adjusts the flow rate according to the ambient pressure. The receiver analyses the intensity of light scattered by aerosols within the airflow, subsequently deriving the aerosol number concentration and size distribution across 31 size bins ranging from 0.253 μm to over 35.15 μm diameters, and also provides results of aerosol mass concentrations (i.e., $PM_{10}$, $PM_4$, $PM_{2.5}$, and $PM_1$). As part of the Aerosol-Cloud Microphysics Monitoring System, a drying tube is employed

to ensure that the relative humidity remains below 40% during sampling. The detection limit of the 11-D for mass concentration is 100,000 μg m$^{-3}$, while for number concentration, it is 3,000 cm$^{-3}$. Ardon-Dryer et al. (2022) reported that the 11-D exhibits an uncertainty of ±3 % when the number concentration exceeds 500 cm$^{-3}$. The $PM_{2.5}$ and $PM_{10}$ measurements from the 11-D were compared with hourly data from the 1405-DF TEOM Continuous Dichotomous Ambient Air Monitor at LABS. The root-mean-square error (RMSE) was 3.76 for $PM_{2.5}$ and 2.97 for $PM_{10}$, with both variables showing a coefficient of determination

($R^2$) of 0.9, indicating good agreement. Therefore, the $PM_{2.5}$ data used in this study are based on the 11-D observations, and the raw data have a temporal resolution of 6 seconds, which were merged into 1-minute averaged data. The aerosol optical properties were analysed using the aerosol system deployed at LABS. The Aerosol system includes a TSI 3563 nephelometer and a Continuous Light Absorption Photometer (CLAP) to investigate aerosol optical characteristics. TSI 3563, which is manufactured by TSI, is capable of measuring the scattering coefficient and backward scattering coefficient for $PM_{10}$ and $PM_1$

at wavelengths of 450 nm, 550 nm, and 700 nm. CLAP is a compact instrument that has been designed by NOAA's Global Monitoring Division (GMD). It is well known for its capacity to conduct continuous long-term monitoring and to measure the absorption coefficients of $PM_{10}$ and $PM_1$ at wavelengths of 467, 528, and 652 nm. By integrating the data from TSI 3563 nephelometer and CLAP, further calculations of aerosol optical properties such as the Absorption Ångstrom Exponent (AAE), Scattering Ångstrom Exponent (SAE), and Single Scattering Albedo (SSA) can be derived for a more comprehensive analysis.

**2.3 Definition of Warm Cloud**

    Gultepe and Isaac (1999) proposed that extremely low number concentrations observed by cloud droplet spectrometers might be attributed to aerosols or dust in the environment rather than to cloud droplets. However, the minimum effective value should be adjusted according to cloud type and in-cloud number concentration to prevent distortion of the average calculation. Over the past two decades, studies utilizing cloud droplet spectrometers have defined the minimum effective value for low-

level clouds as a droplet number concentration of 10 cm$^{-3}$, with liquid water content constrained to a concentration range between 0.001 and 0.01 g m$^{-3}$ based on the expected visibility of either thick or thin cloud layers (Deng et al., 2009; Koike et al., 2019; Yang et al., 2019; D'Alessandro et al., 2021). Deng et al. (2009) conducted research on warm clouds in the East





Asian region and reported an average cloud droplet number concentration of 193 cm$^{-3}$ for the observed middle to low cloud families. This study analyses the aerosol-cloud interactions observed at LABS for low-level warm clouds, defining the warm

cloud data based on previous studies' thresholds for cloud droplet spectrometer sampling (Deng et al., 2009; Yang et al., 2019) as a droplet number concentration of $\geq$ 10 cm$^{-3}$, liquid water content of $\geq$ 0.001 g m$^{-3}$, and an ambient temperature greater than 0°C.

During spring, amounts of biomass burning aerosols from the Southeast Asian Peninsula is lifted into the free atmosphere through the convergence of low-level tropospheric air and transported to East Asia and the western Pacific by a low-level jet

(LLJ) generated by frontal structures at an altitude of 700 hPa (Lin et al., 2013; Yen et al., 2013; Huang et al., 2020). Satellite imagery and surface chart show (Fig. 2 (a), (b)) that the passing of fronts carries abundant moisture from inland China towards East Asia and the western Pacific on March 7, 2024, demonstrating that the location of the cloud system lies north of the biomass burning hotspot, leading to mixing during the transport process. To minimize the variability in aerosol composition and cloud types, satellite imagery and surface charts (Fig. 2 (a), (b)) were used to ensure that the observed warm cloud data

were associated with frontal systems. In addition, 3-day backward trajectories from the NOAA HYSPLIT model (Fig. 2 (c)) were used as one of supporting evidence to identify the potential influence of biomass-burning aerosol transport, indicating that the air masses surrounding LABS during cloud events may have passed through affected regions, such as the southern coastal areas of China and the Southeast Asia Peninsula.

## 2.4 Aerosol-Cloud Interaction Index

In the comparison of the relationship between cloud microphysics and aerosol concentration variations, the Aerosol-Cloud Interaction (ACI) index was first proposed by Kaufman and Fraser (1997) to quantify aerosol effects on cloud microphysics. The ACI index is defined as the ratio of the change in cloud droplet effective radius ($r_e$) to the change in aerosol optical depth (AOD), as expressed in Eq. (1). To minimize observational errors caused by different instruments and locations, Feingold et al. (2001) further revised the formulation into Eq. (2), which has been widely adopted in subsequent studies

(McComiskey et al., 2009; Lihavainen et al., 2010; Zheng et al., 2020; Chen et al., 2021).

$$ACI = \frac{\Delta r_e}{\Delta AOD}, \tag{1}$$

$$ACI = -\frac{AOD}{r_e}\frac{\Delta r_e}{\Delta AOD} = -\frac{d\ln r_e}{d\ln AOD}, \tag{2}$$

Here, $N_d$ is the cloud droplet number concentration, COT is the cloud optical thickness, and LWP is the liquid water path, while $\alpha$ refers to aerosol parameters such as AOD, PM$_{2.5}$, or aerosol number concentration ($N_a$). Based on satellite remote

sensing assumptions such as $COT \propto N_d^{\frac{1}{3}}$ and $r_e \propto LWP/_{COT}$, regarding cloud microphysical variables (Twomey, 1977; Stephens, 1978), the relationship can be extended to Eq. (3) to (5) (McComiskey et al., 2009; Lihavainen et al., 2010).





Therefore, when comparing the ACI index, it is necessary to constrain the analysis to similar LWC conditions to ensure consistency across different parameters. This serves as a constraint when comparing metrics such as cloud effective radius or cloud optical thickness. More intuitively, without fixing LWC, variations in cloud effective radius or optical thickness may reflect intrinsic changes in cloud systems rather than the influence of aerosol concentrations.

$$ACI_{N_d,\alpha} = \frac{1}{3}\left(\frac{\partial \ln N_d}{\partial \ln \alpha}\right),\tag{3}$$

$$ACI_{COT,\alpha} = \left.\frac{\partial \ln COT}{\partial \ln \alpha}\right|_{LWP},\tag{4}$$

$$ACI_{r_e,\alpha} = -\left.\frac{\partial \ln r_e}{\partial \ln \alpha}\right|_{LWP},\tag{5}$$

In contrast, McComiskey et al. (2009) indicated that the cloud number concentration emphasizes the activation process from aerosol to cloud droplet. This metric, particularly when direct measurements of droplet number are available, is less constrained by LWC. While Eq. (3) simplifies the cloud microphysical processes to some extent, it adequately captures the relationship between changes in aerosol number and cloud droplet number concentrations. Theoretically, when all added aerosol particles are activated and grow into cloud droplets, the derivative $\frac{\partial \ln N_d}{\partial \ln \alpha}$ equals 1. Therefore, within the defined range, $0 < ACI \leq 0.33$ represents the reasonable interval for the microphysical process of aerosol particles growing into cloud droplets.

Nevertheless, the variability in atmospheric conditions and the complexity of aerosol composition may lead to ACI index values that exceed the expected range or even become negative. This paper posits that negative ACI index values are not inherently meaningless, nor are they solely influenced by factors external to aerosols, particularly when considering the physical implications of Eq. (3). In certain cases, absorbing aerosols can significantly alter ambient temperature, thereby reducing relative humidity, which makes the environment less conducive to cloud droplet formation (Johnson et al., 2004; Koch and Del Genio, 2010; Boucher et al., 2013; Schultze and Rockel, 2018). More extreme examples of this phenomenon are frequently observed in cloud seeding studies, where competing effects between hygroscopic seeded aerosols and natural aerosols are noted. Since larger cloud condensation nuclei (CCN) require less supersaturation to form droplets, all available moisture tends to accumulate in the larger CCN, forcing smaller CCN to evaporate or remain inactive (Segal et al., 2004; Segal et al., 2007). Whether through direct effects from changes in aerosol number concentrations or indirectly via increased temperature from aerosol absorption, the interplay between the aerosol load and supersaturation can lead to scenarios where an increase in aerosol concentration coincides with a decrease in cloud droplet numbers. When the observed data being compared are based on similar weather conditions and the light-absorbing properties of aerosols are further established, a negative ACI index value has physical significance.





$N_d$, $r_e$, $N_a$, and $PM_{2.5}$ were all derived from in-situ instruments in this study. Given the advantages of $ACI_{N_d}$ metrics, this study primarily adopts Eq. (3) as the basis for calculating the ACI index. Additionally, Sect. 3.4 presents supplementary analyses using $ACI_{r_e}$ (Eq. (5)) calculations under constrained LWC conditions for comparison. However, due to the size range limitations of the 11-D instrument, aerosols smaller than 0.253 μm are not detected, leading to a noticeable underestimation of $N_a$ values. Thus, in calculating ACI index, the variation in $PM_{2.5}$ is primarily used as a proxy for aerosol concentration changes.

**2.5 Cloud-Aerosol Mixing Ratio**

In order to better explain the nonlinear relationship of aerosol concentration and the amount of water content in the cloud system without constraint LWC, it is essential to consider both cloud water content (i.e., LWC, $N_d$) and aerosol concentration (i.e., $PM_{2.5}$, $N_a$) when analysing aerosol effects.

Therefore, this study introduces a ratio-based index to quantify the relationship between cloud water content and aerosol concentration. Drawing on the meteorological concept of mixing ratio, which is the ratio of water vapor to dry air, this study calculates the ratios of measured aerosol and cloud droplet mass and number concentrations to assess aerosol loading in cloud systems relative to liquid water content. Equations (6) and (7) define the mass concentration mixing ratio (MCr) and number concentration mixing ratio (NCr), representing the amount of cloud liquid water per unit aerosol mass and the number of cloud droplets per unit aerosol number concentration, respectively. It is important to note that due to the underestimation of $N_a$ in the 11-D measurements, the study primarily focuses on MCr for quantitative analysis. Furthermore, relative humidity (RH) was not included in the mixing ratio calculations owing to limitations in measurement capabilities. When the ambient RH approaches saturation (~100%), the sensitivity of existing instruments to subtle variations diminishes significantly. However, aerosol activation processes under such near-saturated conditions, such as crossing the critical threshold on the Köhler curve, are highly sensitive to these minor differences (Klemm and Lin, 2016). Not to mention that RH measurements cannot capture supersaturation conditions. While LWC also does not directly quantify supersaturation, it provides a more quantitative representation of the resulting cloud water after a series of processes such as air mass mixing, aerosol activation, and subsequent cloud droplet growth or dissipation.

$$MCr = log\left(\frac{LWC \times 10^6}{PM_{2.5}}\right), \tag{6}$$

$$NCr = log\left(\frac{N_d}{N_a}\right), \tag{7}$$

**2.6 Selection of Time Interval Resolution**

Within a cloud event, the continuous variation in cloud microphysical properties often exhibits strong temporal correlations, effectively capturing the impact of environmental perturbations and indicating whether the increased aerosol




particles have been activated. In addition, to avoid the influence of rapid meteorological changes associated with frontal progression, the Standard Error of Mean (SEM) was used to determine an appropriate temporal scale for analysing ACI index, MCr, and NCr from continuously observed data. SEM illustrates the relationship between the mean calculated from a sample size n and the standard deviation σ of the data within that sample. Wang et al. (2021) used the average SEM (Eq. (8)) to identify an appropriate spatial resolution for observational values. The SEM values of $PM_{2.5}$, LWC, and $N_d$ under different temporal scales were calculated (Fig. 3), ensuring that the differences between different sampling intervals are minimized while also avoiding excessively long-time intervals that could be influenced by significantly different weather conditions.

$$\overline{SEM} = \frac{\overline{\sigma}}{\sqrt{n}}, \tag{8}$$

Figure 3 (a) shows that when the time resolution of the observational data is less than 60 minutes, the average SEM for $PM_{2.5}$, LWC, and $N_d$ is large and decreases rapidly as the time interval increases. However, beyond 60 minutes, the decrease approaches a linear trend, and the differential results exhibit significant variation before and after the 60-minute mark (Fig. 3 (b)). Consequently, this study adopts a resolution of 60 minutes, calculating the moving averages of $PM_{2.5}$, LWC, and $N_d$ for continuous cloud events in 60-minute intervals, while excluding the first 30 minutes and the last 30 minutes of the cloud event for ACI index calculations. Simultaneously, when calculating the moving averages, it is necessary to have a sufficient duration and continuous occurrence of cloud events. Accordingly, in the subsequent analysis of the temporal variations of aerosols and cloud microphysics, the calculations for MCr, NCr, and ACI will be based on data from continuous cloud events lasting more than three hours. The start and end times of each cloud event are presented in Table 2.

## 3 Result and Discussion

### 3.1 Aerosol Optical Properties

Figure 4 shows the aerosol system observations at LABS. High scattering coefficients (Fig. 4 (a)) and absorption coefficients (Fig. 4 (b)) were observed on both March 8 and 9. The daily average scattering coefficient on March 9 was $127 \pm 64$ Mm$^{-1}$, slightly lower than the result on March 8 ($130 \pm 34$ Mm$^{-1}$). However, the absorption coefficient was higher on March 9 ($22 \pm 11$ Mm$^{-1}$) compared to March 8 ($20 \pm 5$ Mm$^{-1}$), indicating that the aerosols on March 9 had stronger light-absorbing properties. Further analysis was conducted using AAE and SAE to evaluate the light absorption and scattering abilities of aerosols at blue and red wavelengths (Fig. 4 (c), (d)). Apart from March 1, where the AAE was significantly lower than 1 ($0.77 \pm 0.12$), and March 12, where it was slightly higher ($1.35 \pm 0.12$), the AAE daily averages remained close to 1, indicating slight variation in the aerosols' absorption across different wavelengths. The SAE remained fairly consistent throughout the observation period, with a clear drop on the morning of March 12, indicating the presence of coarse particles at the station. On March 9, the SAE was slightly lower ($1.49 \pm 0.13$), which could indicate the presence of some coarse aerosols, potentially related to contributions from local sources, especially in the early morning. Furthermore, SSA values were used to assess the





light-absorbing capacity of aerosols (Fig. 4 (e)). During the observation period, most values of SSA were below 0.9, except on March 12. The overall mean was 0.87 ± 0.02, consistent with the values reported in previous studies conducted under the influence of biomass-burning aerosols, which typically ranged from 0.77 to 0.87 depending on transport distance (Hsiao et al.,
2017; Davies et al., 2019; Taylor et al., 2020; Wu et al., 2020). Combining AAE, SAE, and SSA data, and applying the aerosol classification method developed by Cappa et al. (2016) and Schmeisser et al. (2017) for NOAA aerosol systems (Fig. 5), the results suggest that the aerosols observed during the entire study period exhibited optical characteristics typically associated with black carbon type.

Comparing these results with continuous cloud events lasting more than 3 hours (Table 2) shows that pollutant
concentrations gradually increased during the early and late stages of cloud events on March 1 and 7. Pollutant concentrations were higher throughout the cloud event on March 9, whereas March 10 and 11 featured cloud events with lower pollutant concentrations. However, periods with optical properties differing from those of long-range transported absorbing aerosols, such as the coarse-mode aerosols observed on the morning of March 9 and the SSA values exceeding 0.9 after March 12, did not coincide with any cloud events. Therefore, these deviations do not influence the analysis in this study.

**3.2 Cloud Microphysical Characteristics Under Different Aerosol Loads within Cloud Systems**

To compare the impact of aerosol loading on cloud microphysical characteristics within cloud systems, $PM_{2.5}$ values measured concurrently with the warm cloud observation data served as the distinguishing criterion. Due to significant differences between ground-level pollutant concentrations and those at higher altitudes or within clouds, traditional air quality classifications may not effectively differentiate between high and low aerosol loads. Therefore, this research categorized the
data into ten groups, each containing an equal number of data points, to facilitate comparisons while minimizing statistical errors caused by varying dataset sizes.

Figure 6 (a) shows the aerosol loading in the environment, ranging from low (left hand side) to high (right hand side). The average results for $N_d$ reveal that Group 3 exhibited the highest cloud droplet number concentration, averaging at 370 cm$^{-3}$, corresponding to a $PM_{2.5}$ concentration between 1.4 and 2.6 µg m$^{-3}$. The highest LWC (0.219 g m$^{-3}$) was found in Group 2,
with $PM_{2.5}$ concentrations of 0.4 to 1.4 µg m$^{-3}$. Notably, after Group 4, both $N_d$ and LWC exhibited a decreasing trend as environmental aerosol concentrations increased. From Group 1 to Group 5, ED decreased with rising aerosol concentrations; however, beyond Group 5 ($PM2.5 \geq 5.7$ µg m$^{-3}$), ED maintained an average size or slightly increased despite the higher aerosol concentration. A further comparison of weather data (Fig. 6 (b)) across different $PM_{2.5}$ ranges revealed that Group 10 ($PM2.5 \geq 21.7$ µg m$^{-3}$) had the lowest average temperature (4.3°C) and specific humidity (7.0 g kg$^{-1}$), possibly constrained by specific
time and weather conditions. Additionally, relative humidity displayed a decreasing trend with increasing environmental aerosol loading, and the standard deviation of relative humidity increased, reaching the lowest value (93.4%) and the highest standard deviation (7.1%) in Group 9.



The results from the combined analysis of cloud microphysical data and weather data indicate that the variations in aerosols and cloud microphysics described by the Twomey effect (Twomey, 1974, 1977) are more pronounced when $PM_{2.5}$ levels are below 4.2 μg m$^{-3}$ (Group 4). A comparative analysis reveals that Groups 1 to 3 exhibit higher LWC values compared to Groups 6 to 9, highlighting a significant difference in liquid water content within cloud systems under varying aerosol loads. Groups 4 and 5 represent a transitional phase. Groups 1 to 3 illustrate that in an environment with average low aerosol loading, higher LWC coexists with increased $N_d$, while ED decreases, suggesting that sufficient water resources are available to activate more aerosol particles. It is important to note that while LWC does not directly equate to the levels of supersaturation within the cloud system, higher LWC values imply that the initial conditions within the cloud system indeed provided sufficient moisture to support aerosol activation and continued growth. In the high aerosol loading environment of Groups 6 to 9, increased aerosol concentrations result in a decrease in $N_d$ and a slight increase in ED, indicating that cloud droplets are more likely to dissipate, with liquid water becoming more concentrated in larger droplets. Under the classification scheme adopted in this study, it is evident that cloud microphysical characteristics differ significantly when $PM_{2.5}$ concentrations are above or below 4.2 μg m$^{-3}$. In contrast to the relatively consistent trends observed in Groups 1 to 3, Groups 6 to 9 present a distinctly different outcome, demonstrating that the impact of light-absorbing aerosols on cloud systems is not monotonic. Instead, it reveals nonlinear interactions between aerosol loading and cloud microphysical characteristics.

**3.3 Relationship between Cloud-Aerosol Mixing Ratio and ACI Index in Continuous Cloud Events**

The start and end times of continuous cloud events are shown in Table 2, which allows for an analysis of the cloud microphysical responses as aerosol concentrations fluctuate over time. Figure 7 presents MCr values calculated using the average LWC and $PM_{2.5}$ values from each 60-minute period, NCr values derived from $N_d$ and $N_a$, and the ACI index calculated from aerosol and cloud microphysical data within each 60-minute interval (see Sect. 2.6). The results indicate that only approximately 13% (351/2608 data points) of the selected continuous cloud events display a positive ACI index value. When data exceeding the reasonable range (0-0.33) are excluded, all positive ACI index values appear in regions with high MCr (Fig. 7 (a)). This finding suggests that under the influence of high aerosol concentrations, cloud systems tend to adjust by reducing the cloud droplet number concentrations most of the time. The physical mechanism can be understood in the following way. In cloud systems with higher liquid water content, precipitation is often present, leading to enhanced wet deposition of aerosol particles and consequently lower average aerosol concentrations within the clouds. In contrast, when high concentrations of light-absorbing aerosols are present, they tend to warm the environment and reduce relative humidity (see Section 3.5 for details), which suppresses supersaturation and ultimately promotes cloud droplet evaporation, resulting in reduced liquid water content.

Using MCr 4.5 as a threshold (with the minimum MCr value within the reasonable range for ACI index > 0 being 4.64), the proportion of positive ACI values increases from 13% overall to 30% (334/1105 data points) within the MCr > 4.5 range. Further analysis of individual cloud events revealed notable trends. Event 2 showed the most significant variation in both ACI



index and MCr, with the lowest ACI index observed when MCr approached -2.5. A closer examination of data with MCr > 4.5 indicates that 31% (135 data points) showed positive ACI values. Interestingly, when MCr < 2.5, the proportion of aerosols in the environment was notably higher than the liquid water content. Although the ACI shifted from negative to positive as aerosol concentrations increased, due to the limited data (17 data points) and ACI index values exceeding the theoretical range (0-0.33) (McComiskey et al., 2009; Lihavainen et al., 2010), this very high value might have been influenced by different air mass properties rather than solely by aerosol number changes impacting cloud droplet formation. Event 4 occurred in the early morning (Table 2) and exhibited MCr > 4.5, where 93% (148 data points) displayed positive ACI index values. In Events 1 and 5, although the ACI index tended to increase with high MCr values, most of the calculated ACI values remained negative. These events suggest that short-term variations in the cloud system are more strongly influenced by changes in the surrounding environment. However, event 3 occurred almost entirely under conditions of high aerosol concentration and limited liquid water content, resulting in consistently negative ACI index values. The distribution of NCr followed a similar pattern to MCr (Fig. 7 (b)).

After constraining aerosol properties and meteorological conditions, negative ACI index can be meaningfully interpreted (as described in Sect. 2.4). However, most of the previous studies have focused only on positive region. It is important to note that negative values do not have a defined "reasonable range", they simply indicate that an increase in aerosol concentration is accompanied by a decrease in cloud droplets number. The analysis of observational data confirms the relationship between the cloud-aerosol mixing ratio and the aerosol effects on cloud. MCr, NCr, and the ACI index exhibit a positive correlation. Under low cloud water content (low MCr, NCr), the ACI index can drop to as low as -2, reflecting the dissipation of cloud droplets. The critical transition of the sign between MCr and the ACI index discussed in this section further aids in defining the conditions of sufficient or insufficient cloud water content for the statistical analysis of all observational data.

**3.4 MCr and Aerosol Affect Cloud Microphysics**

In Sect.3.3, temporal analysis revealed a positive correlation between the cloud-aerosol mixing ratio and ACI index. To further clarify this relationship and explicitly distinguish conditions with sufficient liquid water content, this section applies various MCr threshold values as a filtering criterion (Fig. 8), aiming to identify the most appropriate division within the full observational dataset. As shown in Fig. 8, the results remain highly consistent regardless of whether $PM_{2.5}$ or $N_a$ is used as the aerosol variable (α) in the ACI calculation. The data above the threshold value are considered indicative of sufficient liquid water content. In this subset and the threshold exceeds 4.5, the ACI values range from 0.08 to 0.12 (Fig. 8 (a)), the correlation coefficients (r) are consistently near 0.6 (Fig. 8 (b)), and the RMSE remains low (Fig. 8 (c)), indicating stable performance. Overall, under conditions of relatively abundant liquid water, the tendency for both aerosol concentration and cloud droplet number to increase appears largely insensitive to variations in the threshold. In comparison, Chen et al. (2021) reported ACI values of 0.09 in suburban areas and 0.06 in polluted regions of northern Taiwan regarding the impact of aerosols on cloud water; McComiskey et al. (2009) found ACI values in California ranging from 0.04 to 0.15; and Lihavainen et al. (2010)





reported ACI values in high-latitude regions of the Northern Hemisphere between 0.01 and 0.17. The ACI calculated in this study is within a reasonable range. Relatively, data below the threshold exhibit poor separation of aerosol influence when MCr < 4.1. It is not until the threshold reaches approximately 4.3 that the lower group shows clearly negative ACI values. These

findings, when combined with previous observations of cloud microphysical trends for aerosol concentration and liquid water content, confirm that an MCr threshold of 4.5 not only retains a greater amount of data within the regime of sufficient liquid water content, but also effectively distinguishes between different data clusters. Given the size and distribution of the observational dataset, the subsequent analysis is conducted using two representative MCr intervals: [3, 4.5] (1,374 data) and [4.5, 6] (1,069 data), which also helps minimize potential meteorological interference under extreme conditions.

Figure 9 further presents the distribution results of $\ln(PM_{2.5})$ and ED, along with the $ACI_{r_e}$ calculated for different LWC ranges after distinguishing based on MCr. The analysis of all observational data indicates that under LWC constraints, $ACI_{r_e}$ ranges between 0 and 0.03 (Fig. 9 (a)). Despite the positive ACI index, the weak correlation coefficients make it difficult to demonstrate that aerosol effects consistently produced the same impact on cloud microphysics. Under the condition where MCr is limited to between 4.5 and 6 (Fig. 9 (b)), it is observed that as $PM_{2.5}$ concentration increases, ED gradually decreases

within the same LWC range. The calculation of $ACI_{r_e}$ shows an improved correlation coefficient, particularly when the LWC ranges from 0.1 to 0.2 g m$^{-3}$, where $ACI_{r_e}$ is 0.08 (r = 0.55). This is comparable to the ACI index derived from $N_d$ under the same constrained MCr condition. The consistency between the calculated results of $ACI_{N_d}$ and $ACI_{r_e}$ suggests that Twomey effect (Twomey, 1974, 1977) is the dominant mechanism within this range.

Conversely, when MCr is within the range of 3 to 4.5 (Fig. 9 (c)), the results from $ACI_{r_e}$ indicate that changes in ED

actually increase with rising $PM_{2.5}$ concentrations. This further supports the notion that the impacts of absorbing aerosols on cloud systems vary with different loading conditions. Under high absorbing aerosol loading, the increase in ED indicates a decrease in small cloud droplets in the environment, while the slight decrease in $N_d$ suggests the dissipation of these small droplets. This outcome could be attributed to the role of absorbing aerosols in raising temperatures and reducing relative humidity by absorbing shortwave radiation (Bond et al., 2013; Haywood et al., 2021; Zhang and Zuidema, 2021). As observed

in Fig. 6, an increase in temperature is also noted in Groups 6-9. The reduction in environmental supersaturation primarily affects smaller cloud droplets, which may fail to sustain growth or even be unable to surpass the Köhler critical point before developing into clouds. Therefore, as aerosol loading transitions from low to high, the microphysical effects of increasing absorbing aerosols on clouds shift from the Twomey effect to semi-direct effect.

**3.5 Difference in Cloud Droplet Size Distribution**

This section explores the differences in cloud droplet characteristics under high and low aerosol concentrations within different MCr ranges, and conducts comprehensive analysis. Figure 10 displays the frequency of cloud microphysical features based on MCr grouping. By dividing each group into high and low aerosol loading based on the median value, it was observed





that under high aerosol concentrations in the MCr [4.5, 6] range, the frequency of cloud droplet number concentration within 300-600 cm⁻³ is notably higher than in the results for all observations and the low aerosol load in MCr [4.5, 6] (Fig. 10 (c)). In

the frequency distribution of ED (Fig. 10 (b)), it was also found that frequencies below 12 μm are significantly higher, which is consistent with the ACI index analysis results. In the MCr [3, 4.5] range with high aerosol load, the difference in cloud droplet number concentration is not as pronounced (Fig. 10 (f)); however, the ED results indicate that droplets smaller than 10 μm diameters are more frequent under low aerosol load (Fig. 10 (e)), while the high aerosol load resembles the overall observation results.

The frequency distribution of LWC indicates that within the constrained MCr [4.5, 6], the occurrence of LWC values exceeding 0.15 g m⁻³ is noticeably higher (Fig. 10(a)). Conversely, in the MCr [3, 4.5], cloud events with lower LWC appear more frequently, especially under low aerosol loading conditions (Fig. 10 (d)). A comparison of the frequency distributions of ED and $N_d$ reveals that variations in droplet size, rather than droplet number concentration, primarily account for the differences in LWC across these regimes. However, it is worth noting that in comparing the effect of aerosol concentration on

ED, it is necessary to restrict the LWC range to reduce uncertainties. Considering weather conditions, Table 3 shows the mean value of temperature, relative humidity, specific humidity in different constraint conditions. In the high aerosol pollution range of MCr [3, 4.5], the temperature is higher, the moisture content is greater, but the relative humidity is lower, aligning with the description of the semi-direct effect. On the other hand, the lower specific humidity observed under low aerosol loading suggests a reduced availability of water vapor, which may be one of the contributing factors to the smaller cloud droplet sizes.

Figure 11 presents the average distributions of aerosol and cloud droplet sizes, while Figure 12 further illustrates observations within the LWC range of 0.1-0.2 g m⁻³. From the comparison chart for MCr [4.5, 6] (Fig. 11 (a)), it is observed that aerosol particles are primarily maintained below 5 μm, with an overall lower concentration. Under high aerosol load, there is an apparent increase in smaller cloud droplets. However, as shown in Fig. 10(a), higher aerosol loading is often associated with greater liquid water content (LWC), with elevated average LWC also observed (0.25 and 0.19 g m⁻³ are the means in high

and low aerosol load). Likewise, more cloud droplets larger than 20 μm are observed under high aerosol load. To further minimize systematic interference and better align with the assumptions of the Twomey effect, applying an LWC constraint is both necessary and appropriate (Fig. 12 (a)). When LWC is restricted (Fig. 12 (a)), the aerosol size distribution remains similar to the unconstrained case (Fig. 11 (a)), except for the reduction in particles larger than 2.5 μm. Under high aerosol loading, a significant increase in cloud droplets smaller than 12 μm is observed, while the larger droplets (> 12 μm) markedly decreases.

This shift differs from the case without LWC constraints. It more clearly characterizes the Twomey effect, illustrating how aerosol particle concentrations influence the droplet size distribution, in particular, there are minimal differences between high and low aerosol pollution in terms of mean temperature (5.8°C for both), relative humidity (98% for both), and specific humidity (7.83 g kg⁻¹ vs. 7.81 g kg⁻¹) (Table 3).



Within the MCr range of [3, 4.5], Figure 11 (b) first reveals that under low aerosol loading, only fine-mode aerosol
particles are present. Examining the cloud droplet size distribution, with 10 μm as the threshold, more droplets smaller than
10 μm are observed under low aerosol loading, while the number of droplets larger than 10 μm sharply decreases. Notably, the
total $N_d$ under low aerosol loading reaches 223 cm$^{-3}$, slightly higher than the 217 cm$^{-3}$ observed under high aerosol loading.
Compared with the observed mean (Fig. 11 (b), black solid line), cloud droplets smaller than 10 μm under low aerosol loading
exceeds the average. In contrast, for cloud droplets larger than 10 μm, the number remains below the mean regardless of aerosol
loading. The differences from the original conditions become less pronounced when LWC is constrained (Fig. 12 (b)).
Focusing on high aerosol loading results, the number of small cloud droplets shows lower than in both average (the black line
represents the observation average for LWC 0.1-0.2 g m$^{-3}$) and low aerosol loading cases. On the other hand, under low aerosol
loading, the number of large droplets (> 12 μm) is less compared to the average and high aerosol loading scenarios. Overall,
around the 10 μm threshold remains a clear dividing point. Combined with the observation of larger coarse-mode aerosol
particles in high aerosol loading conditions, this suggests that coarse-mode aerosols can activate and grow into larger droplets
under lower supersaturation conditions, but small droplets are more prone to dissipation, consistent with their position on the
Köhler curve (Klemm and Lin, 2016). In terms of weather conditions, under constrained LWC, the high aerosol pollution
interval in MCr [3, 4.5] has a mean temperature of 6.8°C, relative humidity of 94%, and specific humidity of 8.02 g kg$^{-1}$. Under
low aerosol pollution, the mean temperature is 5.8°C, relative humidity is 98%, and specific humidity is 7.80 g kg$^{-1}$. Differences
in these results agree with the previous discussion (Table 3) in this section.

A comprehensive comparison under different cloud-aerosol mixing ratios, aerosol loadings, and meteorological
conditions reveals that in the MCr range of [4.5, 6], the Twomey effect can be clearly and consistently described through
changes in the cloud droplet size distribution. In contrast, within the MCr range of [3, 4.5], the interpretation becomes more
complex. Based on the Köhler curve, coarse-mode aerosols under high aerosol loading can more easily activate and grow into
larger droplets under lower supersaturation conditions, while smaller droplets are more prone to evaporation, which is
consistent with the observed cloud droplet size distributions. Although this study does not directly measure supersaturation
levels, the observed results—characterized by higher temperatures and lower relative humidity—align with the theoretical
framework of the semi-direct effect.

## 4 Conclusion

This study applies the cloud-aerosol mixing ratio (MCr and NCr) to systematically evaluate the interplay between cloud
liquid water and aerosol content.  In particular, these ratios quantify the amount of liquid water per unit mass of aerosol and
provide a more integrated perspective for distinguishing microphysical responses under different environmental conditions,
highlighting when aerosols are more likely to be activated or when cloud droplets tend to be suppressed. Based on cloud and
aerosol microphysical data collected in March 2024 at the Lulin Atmospheric Background Station (LABS) in central Taiwan,





we identify a nonlinear relationship between aerosol concentration and cloud microphysical properties. In a comparative analysis involving 10 data groups stratified by both $PM_{2.5}$ concentration levels and sample size considerations, a distinct trend emerged: in the low-concentration groups (Groups 1-3), an increase in $PM_{2.5}$ was associated with an increase in $N_d$ and a significant decrease in ED. However, in the high-concentration groups starting from Group 6 ($PM_{2.5} \geq 5.7$ µg m$^{-3}$), further increases in $PM_{2.5}$ corresponded to a decrease in $N_d$ and a slight increase in ED. These results suggest a transition in cloud

microphysical responses under elevated aerosol loading conditions.

    Building upon the nonlinear response, the subsequent analysis examined the association between the ACI index and the cloud-aerosol mixing ratio over a continuous 60-minute period. The results indicate that when the cloud system exhibits a higher mixing ratio, a continuous increase in aerosol concentration tends to result in a higher $N_d$. This trend is consistently observed in both MCr and NCr analyses. According to the distribution of the ACI index concerning MCr, only 30% of the

data exhibit ACI > 0 in the MCr > 4.5 regime, remaining within a physically reasonable range. This indicates that the environmental changes in temperature and humidity induced by the absorbing aerosol components often outweigh their potential to activate as CCN. Subsequent analyses based on varying MCr thresholds further confirm the appropriateness of selecting MCr = 4.5 as a critical value in this study. A further examination of effective diameter showed that under constrained LWC conditions, the trends of effective diameter with aerosol concentration differed significantly between the MCr [4.5, 6]

and [3, 4.5] regimes, as reflected by the $ACI_{r_e}$ index. This provides further evidence that the mixing ratio of cloud water to aerosol concentration influences the dominant microphysical processes within the cloud system. Under conditions with sufficient liquid water content (MCr [4.5, 6]), the Twomey effect (Twomey, 1974, 1977) plays an important role.

    In the subsequent analysis, differences between high and low aerosol loads within different cloud-aerosol mixing ratio intervals were examined using MCr. Under MCr [4.5, 6], cloud systems more frequently exhibited higher $N_d$ (300-600 cm$^{-3}$)

and smaller ED (8-12 µm). Regardless of whether LWC was constrained, the cloud droplet size distributions were more concentrated below 10 µm. These results support and reinforce the previous argument. In contrast, under the MCr [3, 4.5] condition, markedly different cloud microphysical characteristics were observed. At low aerosol loading, ED was often smaller than 10 µm, with a greater number of small droplets. At high aerosol loading, more coarse-mode aerosols were present, and the droplet size distribution showed a significant reduction in the number of droplets smaller than 10 µm. These subtle

differences could be explained by Köhler theory. Environmental changes induced by absorbing aerosols lead to a decrease in supersaturation, making smaller aerosols less likely to activate into cloud droplets and existing small droplets more susceptible to evaporation.

    Notably, all of the above analyses are established by simultaneously accounting for both cloud water content and aerosol concentration. Without the additional information provided by MCr or NCr, such variations might be obscured or even

misinterpreted due to mutual interference between variables. The conclusions drawn in this study demonstrate the importance



of incorporating cloud-aerosol mixing ratios—particularly when attempting to transition from observational phenomena to a more complete framework for understanding the nonlinear responses between aerosol concentrations and clouds.

## Data availability

The in-situ data collected by the aerosol–cloud microphysics monitoring system at LABS can be requested from the corresponding author. Supporting data from LABS are available at https://data.moenv.gov.tw/dataset/detail/AQX_P_26 or can also be requested by contacting the corresponding author.

## Author contribution

PHL drafted the initial manuscript and was responsible for formal analysis, software implementation, and data visualization. SHW provided the methodology, resources, and supervision, and contributed to manuscript review and editing. PHL, SHW, OK, and NHL jointly interpreted and discussed the results. All authors contributed to the final paper.

## Competing interests

The authors declare that they have no conflict of interest.

## Acknowledgements

This research is primarily based on the first author's M.Sc. thesis completed at the Department of Atmospheric Sciences, National Central University, Taiwan, and has been further expanded through additional analyses and discussions. The authors would like to express their sincere gratitude to Prof. Sheng-Hsiang Wang for the valuable guidance during the thesis period and constructive suggestions for this research. This research was supported by the National Science and Technology Council (NSTC), Taiwan, under Grant No. MOST 111-2628-M-008-002-MY3. The authors also acknowledge the Ministry of Environment (MOE), Taiwan, for providing access to the Lulin Atmospheric Background Station (LABS) and for supplying air quality and meteorological data.

## Financial support

This research has been supported by the National Science and Technology Council (NSTC), Taiwan (No. MOST 111-2628-M-008-002-MY3).



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






**Table 1. Parameters Used in This Study.**

|  | Parameters | Acronym | Unit | Time Resolution |
|---|---|---|---|---|
| Lulin Weather System | Temperature | T | °C | 1 min |
|  | Pressure | P | hPa |  |
|  | Relative Humidity | RH | -- |  |
|  | Wind Speed | WS | $m\ s^{-1}$ |  |
|  | Wind Direction | WD | degree |  |
|  | Visibility | -- | km |  |
| Lulin Aerosol System | Absorption Coefficient | -- | $Mm^{-1}$ | 1 hour |
|  | Scattering Coefficient | -- | $Mm^{-1}$ |  |
|  | Absorption Ångström Exponent | AAE | -- |  |
|  | Scattering Ångström Exponent | SAE | -- |  |
|  | Single Scattering Albedo | SSA | -- |  |
| Aerosol-Cloud Microphysics Monitoring System | CDP | Cloud droplet number concentration | $N_d$ | $cm^{-3}$ | 1 min |
|  |  | Cloud Number Size Distribution | CSD | $cm^{-3}$ |  |
|  |  | Effective Diameter | ED | µm |  |
|  |  | Liquid Water Content | LWC | $g\ m^{-3}$ |  |
|  | 11-D | Aerosol Number Concentration | $N_a$ | $cm^{-3}$ | 1 min |
|  |  | Aerosol Number Size Distribution | ASD | $cm^{-3}$ |  |
|  |  | $PM_{2.5}$ | -- | $µg\ m^{-3}$ |  |






**Table 2: the start and end local times of continuous cloud events (UTC +8).**

| Cloud Event | Start Time (LT) | End Time (LT) |
|---|---|---|
| Event 1 | 3/1 09:37 | 3/1 15:40 |
| Event 2 | 3/7 09:16 | 3/8 07:28 |
| Event 3 | 3/9 11:07 | 3/9 15:27 |
| Event 4 | 3/10 02:13 | 3/10 06:00 |
| Event 5 | 3/11 09:47 | 3/12 05:21 |

**Table 3: Mean temperature and humidity across different conditions.**

| | | Temperature (°C) | RH (%) | Specific Humidity (g kg$^{-1}$) |
|---|---|---|---|---|
| **MCr [4.5, 6]** | High aerosol loading | 5.8 | 98 | 7.85 |
| | Low aerosol loading | 5.6 | 98 | 7.76 |
| **MCr [4.5, 6] in LWC [0.1, 0.2]** | High aerosol loading | 5.8 | 98 | 7.83 |
| | Low aerosol loading | 5.8 | 98 | 7.81 |
| **MCr [3, 4.5]** | High aerosol loading | 6.4 | 95 | 7.92 |
| | Low aerosol loading | 5.5 | 97 | 7.62 |
| **MCr [3, 4.5] in LWC [0.1, 0.2]** | High aerosol loading | 6.8 | 94 | 8.02 |
| | Low aerosol loading | 5.8 | 98 | 7.80 |




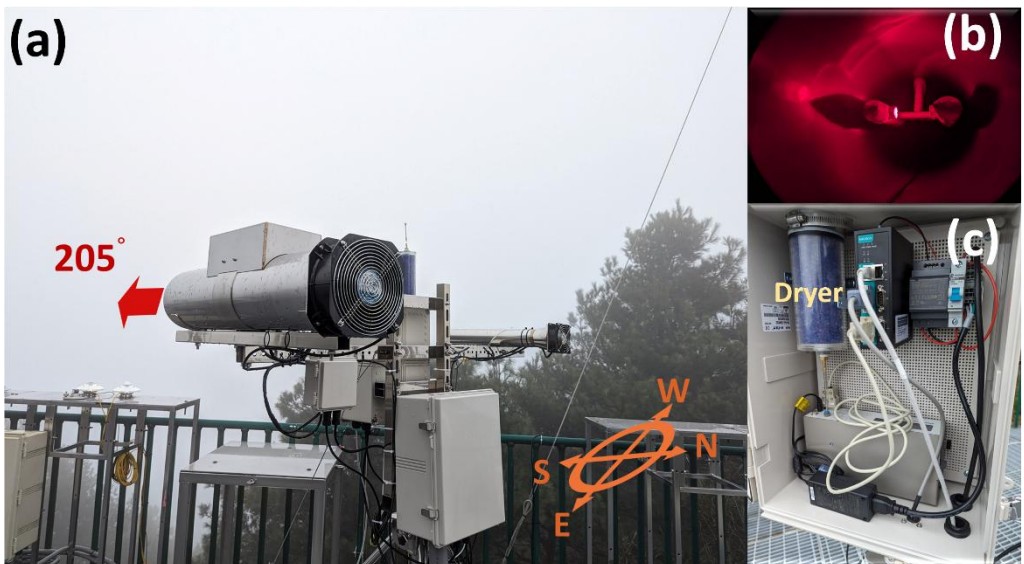

**Figure 1: (a) The setup ang Orientation diagram of Aerosol-Cloud Microphysics Monitoring System deployed at LABS during the 2024 observation period. (b) The appearance of CDP. (c) The appearance of 11-D.**



**Figure 2: (a) Himawari 8 true color satellite image and (b) surface chart from Central Weather Administration (CWA) at 08 LT (UTC +8), March 7, 2024. (c) 72-hour backward trajectories were calculated using the NOAA HYSPLIT model (GFS 0.25° × 0.25°). The trajectories correspond to the main cloud event time (DD_HH, LT) and are shown at 3-hour intervals.**





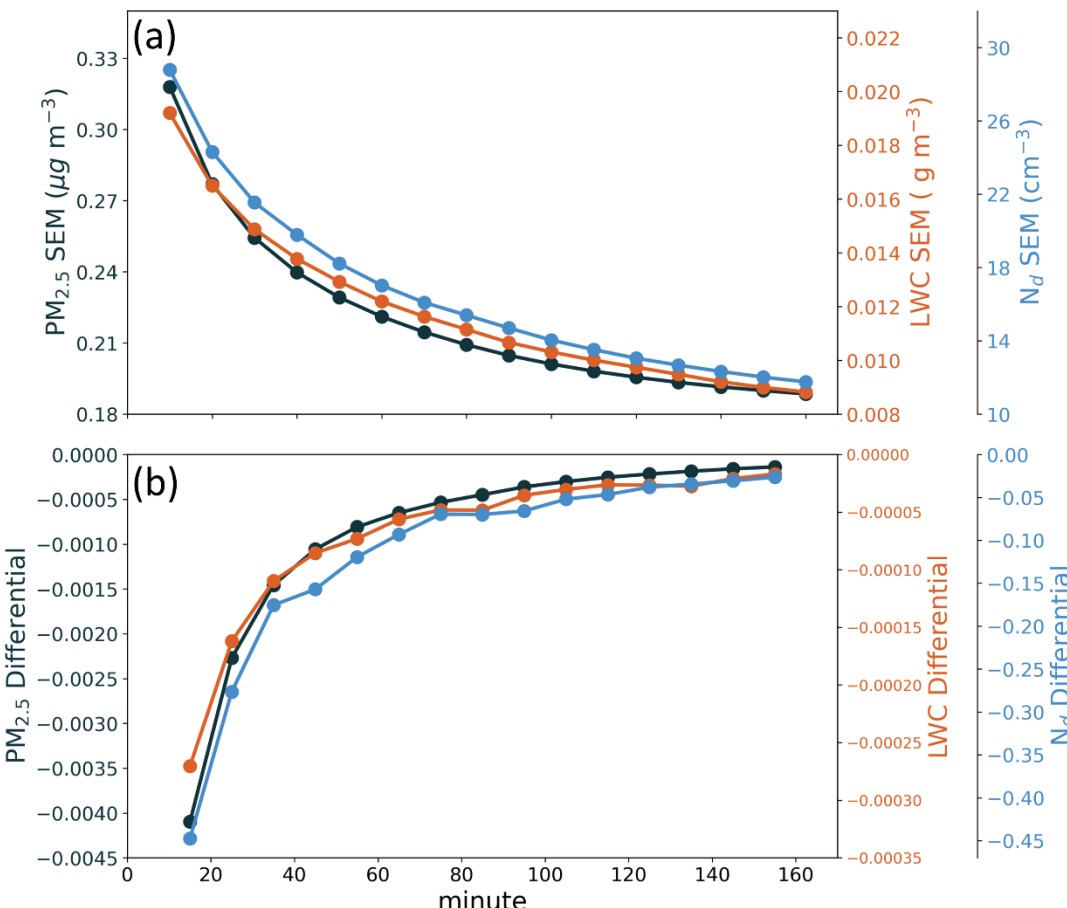

**Figure 3: (a) the standard error of the mean for PM₂.₅ (black), LWC (orange), and N_d (blue) at different time resolutions; (b) the differential results of the standard error of the mean.**





**Figure 4: PM$_{10}$ aerosol optical properties observed by the aerosol system at LABS. (a) absorption coefficient; (b) scattering coefficient; (c) Absorption Ångström Exponent (AAE, blue) and Scattering Ångström Exponent (SAE, orange); (d) Single Scattering Albedo (SSA), and gray dash line means SSA = 0.9.**





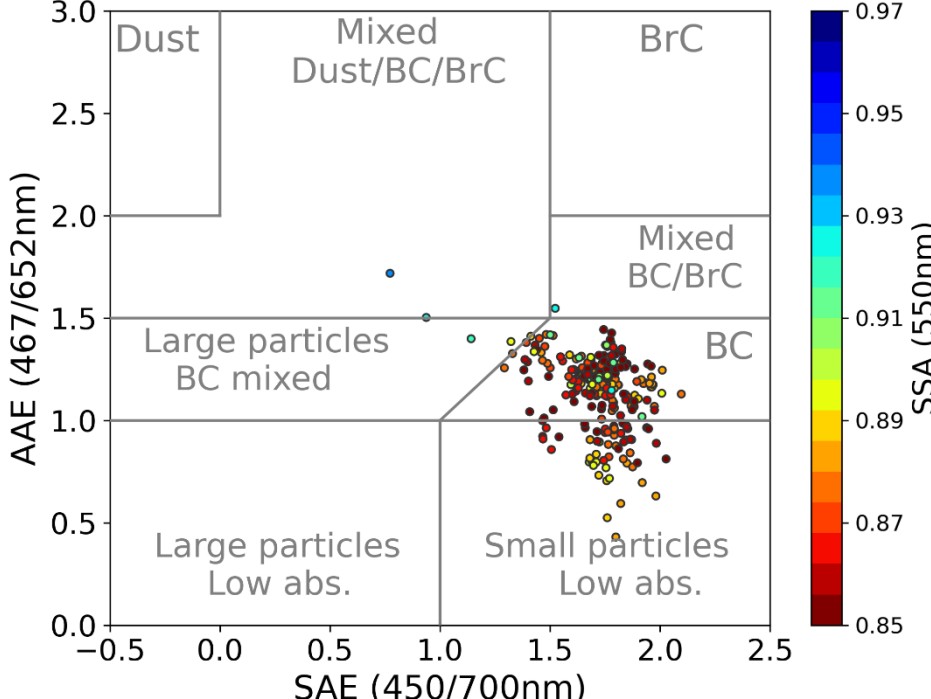

**Figure 5: Classification of aerosols at LABS from March 1 to March 12, 2024 (Classification method adapted from Schmeisser et al.**
**(2017)), BC means black carbon type, BrC means brown carbon type.**



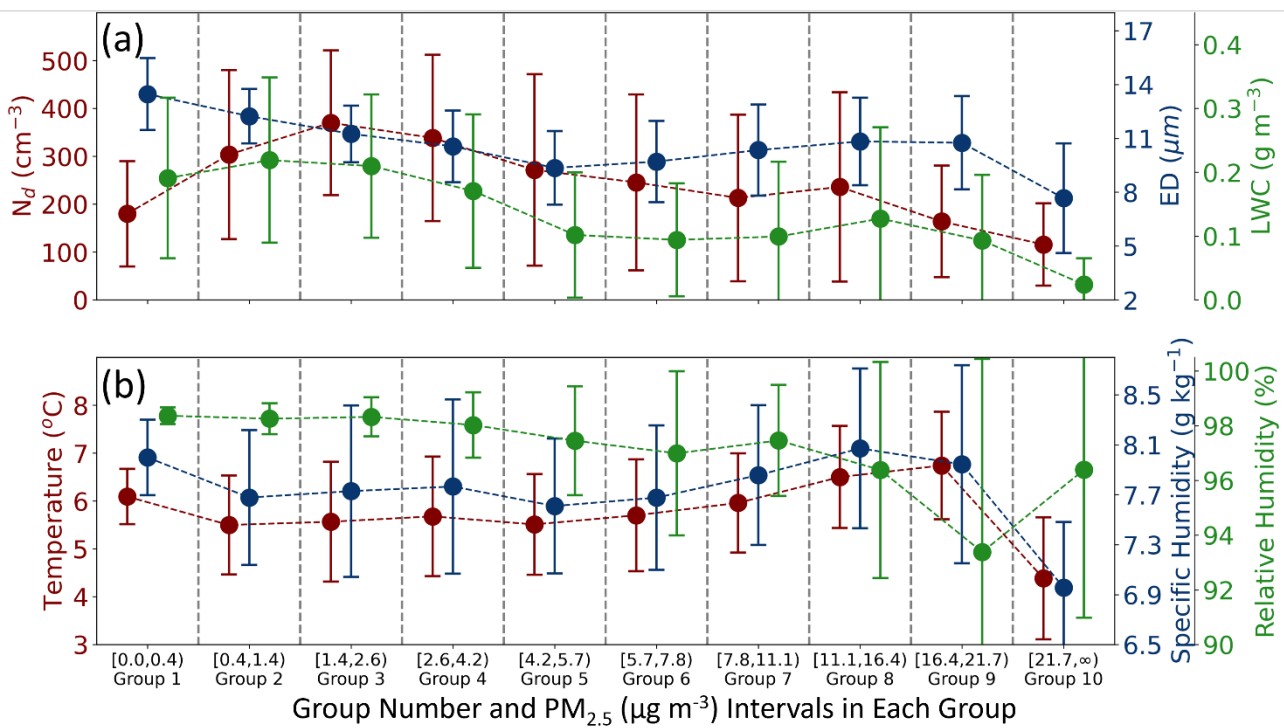

**Figure 6: Cloud microphysical and weather data categorized into 10 groups of equal sample size based on PM$_{2.5}$ obtained from 11-D, ranging from low to high environmental aerosol loading from left to right. (a) CDP observation results, including N$_d$ (red), ED (blue), and LWC (green); (b) Weather results measured at LABS, including temperature (red), relative humidity (green), and specific humidity (blue).**

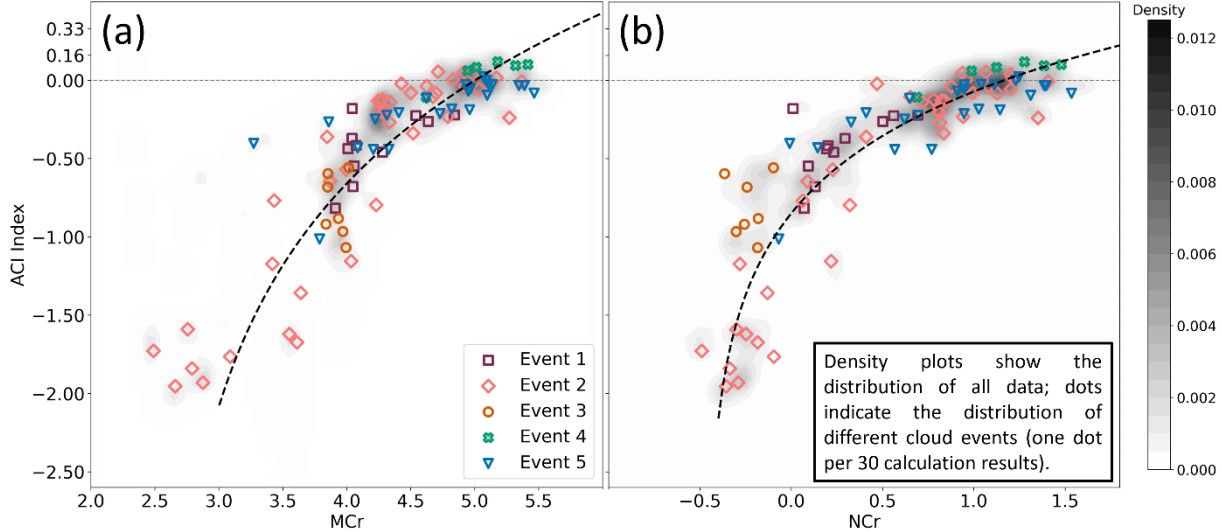

**Figure 7: Scatter and density plots for continuous cloud events: (a) MCr versus ACI index; (b) NCr versus ACI index.**





**Figure 8: (a) ACI values, (b) correlation coefficients, and (c) RMSE are calculated based on different MCr thresholds. Blue markers represent data subsets with MCr values above the given threshold, while red markers indicate those below it. Circles correspond to calculations using PM$_{2.5}$ as α, and triangles denote those using N$_a$ as α.**



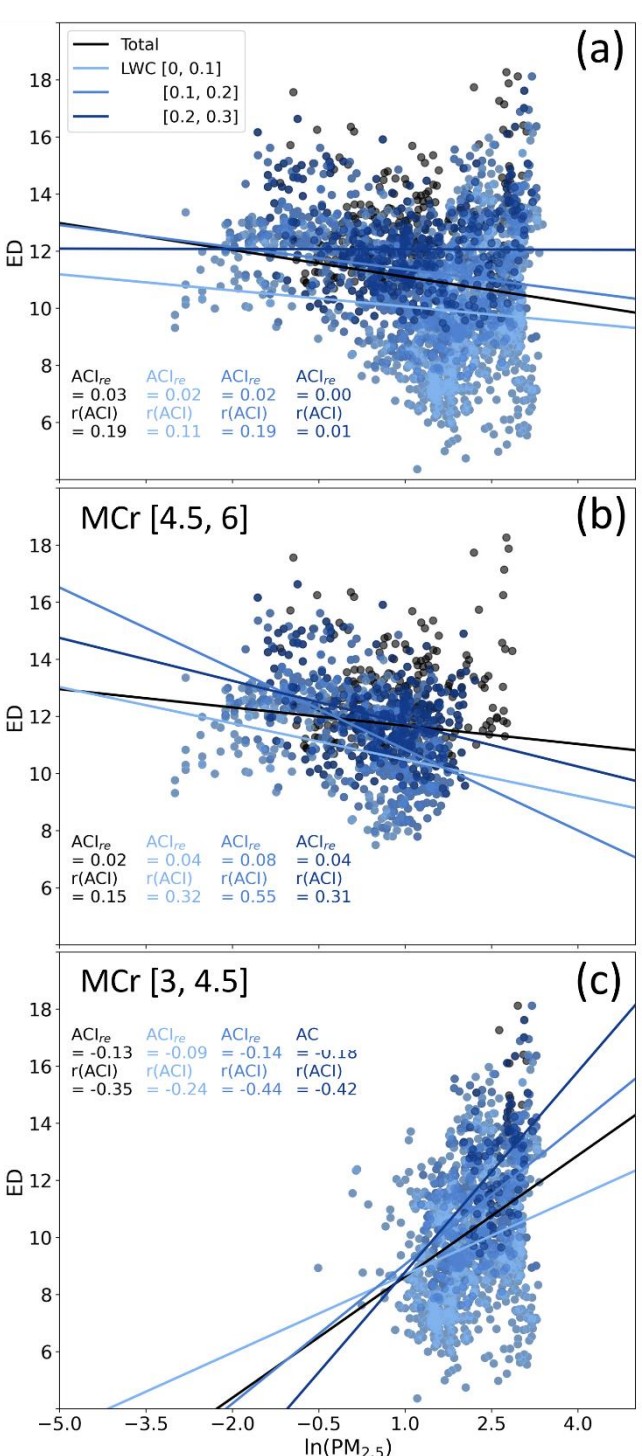

**Figure 9: Scatter plot of ln(PM$_{2.5}$) and ED, along with ACI$_{re}$ calculation results under specified LWC conditions during the observation period. (a) Results for all observed data; (b) MCr [4.5, 6]; (c) MCr [3, 4.5].**



**Figure 10: Frequency of cloud microphysical data during the observation period and within specific MCr ranges. The red line represents high aerosol load conditions, where PM2.5 values exceed the median after MCr filtering; the green line represents low aerosol load conditions, where PM2.5 values are below the median after MCr filtering. (a), (b), and (c) respectively show the frequency of LWC, ED, and Nd data within the MCr range of [4.5, 6]; (d), (e), and (f) show the frequency of LWC, ED, and Nd data within the MCr range of [3, 4.5].**



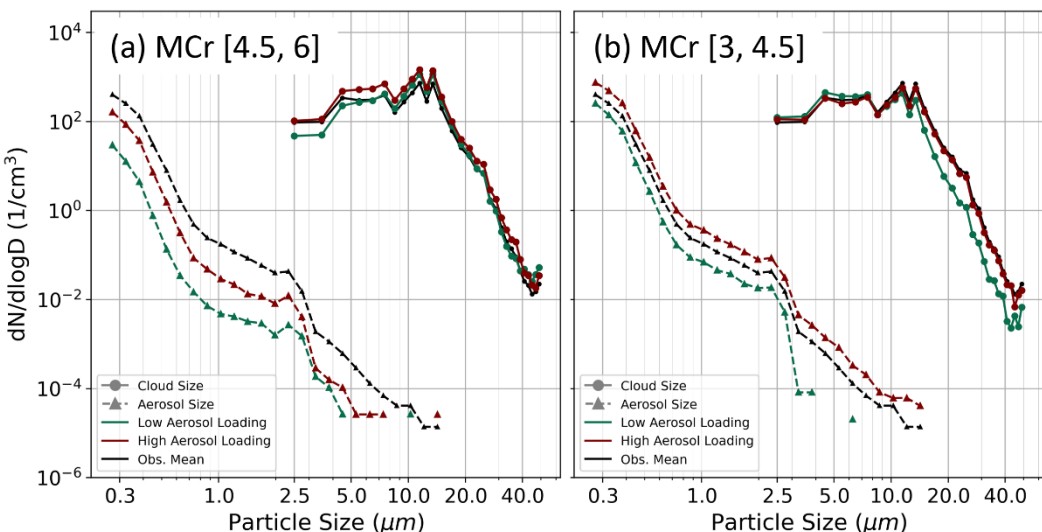

**Figure 11: Size distribution of aerosols (triangles) and cloud droplets (circles), with the black line representing the observation period average. Red and green lines follow the same conditions as in Figure 10. (a) Red and green lines limited to the MCr range [4.5, 6]; (b) Red and green lines limited to the MCr range [3, 4.5].**

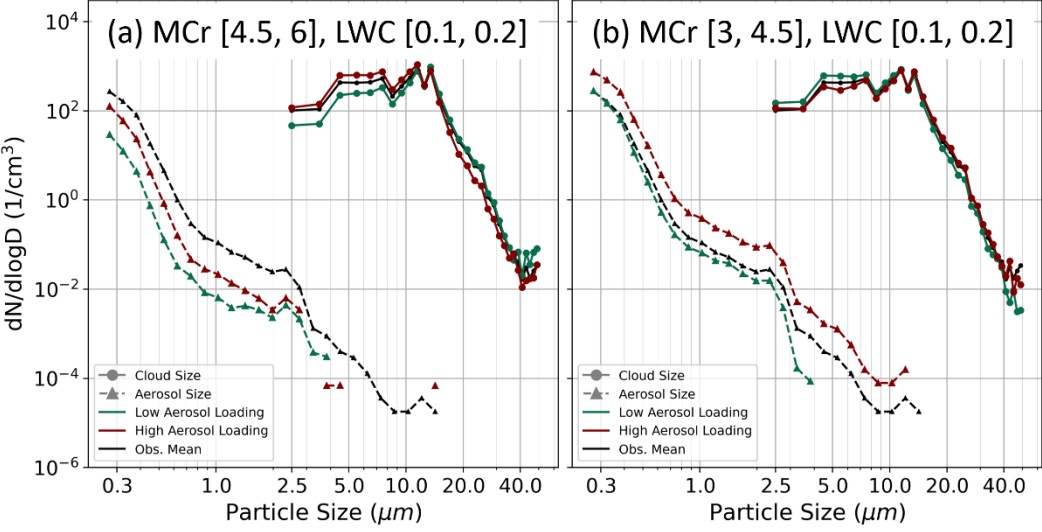

770

**Figure 12: Same as Fig. 11, but all data are limited to the LWC range of 0.1-0.2 g m⁻³.**