# Peer review of "Assessing Nonlinear Responses of Low-Level Warm Clouds Under the Impacts of Absorbing Aerosols Using the Cloud-Aerosol Mixing Ratio"

_EGUsphere, 2025_

## Referee Comment (RC1)

Review of "Assessing Nonlinear Responses of Low-Level Warm Clouds Under the Impacts of Absorbing Aerosols Using the Cloud-Aerosol Mixing Ratio" (Manuscript ID: egusphere-2025-3777)

**Summary**

The research aims to quantify cloud microphysical responses to different mixing ratios of aerosol mass/number to cloud liquid water content using data collected in Taiwan during episodes characterized by absorbing aerosol. The authors address the question: How do cloud microphysical properties respond to a range of mass/number ratios between cloud water and aerosols (MCr/NCr)? The science question is indeed important to address uncertainties in the relationships between aerosols and clouds under different cloud water-aerosol mixing states. The authors introduce a novel quantity (MCr, NCr) and demonstrate its applicability to diagnosing aerosol-cloud relationships, specifically the Twomey and semi-direct effects. The data are relevant for addressing their science question. The figures and tables support the key points of the manuscript but require revision for improved clarity/communication. Some parts of the analysis and writing should be improved as described below. I recommend major revisions prior to consideration of publication.

**Major Comments**

- MCr = 4.5 as a transition point for Twomey and semi-direct effects: Is this valid only for absorbing aerosols? I think there needs to be a control case where the ACI-MCr analysis is applied to a period characterized by scattering/non-absorbing aerosols (e.g., background, non-biomass burning (BB) air) to demonstrate that the semi-direct effect the authors observed here is strongly linked to the presence of absorbing aerosols. Since identifying the effect of absorbing aerosols is a central goal of the study, I believe a control case would be a clear demonstration that the study's findings are indeed related to the semi-direct effect.

- Regarding the possibility of "different air mass properties" (L349), the authors must demonstrate that the observed cloud responses are mainly due to the aerosol-cloud mass ratio (MCr) and not shifting air mass sources/aerosol properties over the sampling period (e.g., shift between aerosols from local vs regional, urban vs BB). As an example, Fig 6 shows the observed response of cloud microphysics grouped by PM2.5 loading. But for a causal relationship to be established, Fig 6 implies consistent aerosol source/composition/properties across different PM2.5 loadings, which the authors must demonstrate. The authors must constrain this confounding variable (aerosol composition/properties) to draw more robust conclusions about the relationship between cloud microphysics and PM2.5 loading. Alternatively, the authors can demonstrate that aerosol composition/properties are roughly consistent across PM2.5 loadings. To do this, I suggest plotting PM10 aerosol optical properties from the Lulin Aerosol System grouped by 11-D PM2.5 loading. Although this is not a one-to-one comparison, this may at least suggest that there is some degree of consistency in PM properties vs. loading.

- Washout of BB aerosol in frontal system: Won't precipitation in frontal system scavenge the biomass burning aerosols? How do the authors know that the aerosols they're sampling aren't local (e.g., Kaohsiung)? While Fig 2 show trajectories reaching the Southeast Asian Peninsula, these trajectories do not preclude the possibility of local influence and do not imply that the BB aerosols were unscavenged during the 2-3 days of transport in the warm front. The authors

must either demonstrate that the source of the absorbing aerosol is Southeast Asian biomass burning or rephrase the text on the origin of the absorbing aerosol.

- I strongly recommend adding a conceptual aerosol-cloud figure to visually explain the key findings of the paper, which can be used as a graphical abstract as well.

**Minor Comments**

- L11-12/L81-86/L106-108: The trajectories from Fig. 2c appear to mainly pass through southeast China and the surrounding waters around Taiwan. The trajectories that do reach the Southeast Asian Peninsula are quite few (only two). The specific source of the absorbing aerosol should not affect the conclusions of the paper, but since Southeast Asian biomass burning was mentioned several times, more evidence that the aerosols originated from biomass burning in that region rather than southeast China is required. For example, the authors could add hotspots over the period of the backtrajectories in Fig. 2c to reinforce claim over the source of absorbing aerosols.

- L80-81: The sentence "Observational studies have…" should be moved to the paragraph before this one for better organization.
- L102: Please mention if the aerosol measurements at LABS have a drying tube for in-cloud periods. Please also specify if the optical properties in Fig 4 refer to dried or ambient aerosols.
- L129: Please specify if the 11-D aerosol measurement during cloud events refers to interstitial aerosol and/or the in-droplet aerosol.
- L237: Please justify the use of PM2.5 over PM10 in the formula for MCr.
- L238: Please clarify what aerosol size range N_a refers to.
- Sect 3.1: It would be helpful to the reader if the authors explicitly stated in this section that the aerosol optical properties refer to PM10 while the rest of the paper refers to PM2.5.
- L260/Sect 3.1: The purpose of Sect 3.1 (PM10 optical properties) needs to be explicitly/more clearly explained in the context of the rest of the paper which uses PM2.5 and focuses only on cloud events. I suggest reframing Sect 3.1 to introduce the cloud events (Table 2) and then describe PM10 aerosol properties before, during, and after each cloud event (Fig 4). This would then flow more smoothly into Sect 3.2. If the LABS optical data is valid in-cloud (e.g., if a dryer was placed upstream of the instruments), it would be informative to include this information as additional columns in Table 2 and added to the discussion in this section.
- L260/Sect 3.1: I recommend adding a meteorological time series to add context to Fig 4 and Sect 3.1 (with similar markings/shadings to show when cloud events occurred). This would help establish a stronger link between the key findings and the physical environment during the sampling period. This meteorological time series could be in the supplementary information.

- L285/Section 3.2: This section is discussed in terms of Group Number. To keep the discussion physically tied to the atmosphere, I suggest that the discussion be based on the PM2.5 concentration ranges (physically based) rather than group number (statistically based). Please rephrase this section so that Group Numbers are replaced by PM2.5 concentration ranges. This will also make the section more succinct. Please apply this change to other sections that mention the PM groups too (e.g., L463).

- L296: Please define ED here or sooner if it's mentioned.

- L318/Sect 3.3: This section requires major revision for clarity, specifically more explicitly connecting the conclusions in the text to the figures so it will be easier for the reader to understand how the data supports the authors' conclusions. See my comments below.

- L336-340: it is not clear in any figure that the PM2.5 > LWC when MCr < 2.5 (since MCr is a ratio, a drop in MCr could be due to changes in LWC and/or PM2.5 and MCr < 2.5 does not necessarily imply that PM2.5 > LWC). It is also not evident from Fig 7 that ACI shifts from negative to positive as aerosol concentrations increase, please provide some x/y values to orient the reader to which part of Fig 7 to refer to. The panels in Fig. 7 are not clearly referenced in the text (i.e., only Fig. 7 is mentioned so it is unclear which panel to refer to while reading the text).

- L344-345: It is not evident from the figures that Event 3 occurred under high aerosol concentrations and low LWC. To add physical context to each cloud event, please add new columns to Table 2 with statistics on cloud microphysical and aerosol properties (e.g., mean+STD PM2.5) for each event. This would help with contextualizing the differences discussed in Sect 3.3 and explaining differences in how each event fills the 2D space in Fig. 7.

- L357: "sufficient liquid water content" please clarify: sufficient for what? For the MCr ratio to be applicable? Or for the Twomey effect to be dominant?

- L358: "division" please clarify what this division is for. If this division/threshold of MCr refers to the point where Twomey or semi-direct effects dominate, please specify this explicitly to make the purpose of Sect 3.4 clearer.

- L375-383: Please state the implication/importance behind why ED does not strongly correlate with ln(PM2.5). Please also justify the use of ln(PM2.5) rather than PM2.5.

- L386-388: Can this conclusion about the droplet size distribution be seen in Fig. 11-12? If so, please mention that to support this conclusion.

- L454/Sect 4: Please describe some suggestions for future analysis based on the main conclusions of the manuscript.

- Fig 4: Please mark each cloud event from Table 2 (e.g., grey areas spanning the start until end of each event). Also mention in caption that daily averages and STDs are also shown.

- Fig 5: please mention the PM size range for these measurements in the caption.

- Fig 6: please mention in the caption how many samples are in each group.

- Fig 6: Because of the uneven width of each group (a consequence of the equal-number groupings), I suggest the x-axis should be PM2.5 concentration rather than group number. This will show the true scale of the x-axis to make it clear that the observed trends are most conclusive for PM2.5 < 10 micrograms m-3 (where most data are). For clarity, one variable can take up one panel (so 6 panels).

- Fig 6: Instead of infinity, Group 10 should list the max PM2.5 concentration recorded during the period.

- Fig 8: Please include a panel to show the number of data points included in the calculation of the y-axis per threshold value (x-axis) to demonstrate that the converging trends mentioned are robust.

- Fig 8b: Please explicitly state in the caption what two quantities the correlation is being calculated for (e.g., ACI and MCr for data where MCr > threshold?). Please apply this to Fig. 9 too perhaps by changing the notation of r(ACI) (e.g., r(ACI, MCr)) for clarity.

- Fig 9: For visual clarity, please use different colors (rather than shades of blue) for different LWC groupings.

- Fig 11: Please double check the coloring of the aerosol size distributions in (a). The observation mean size distribution is not between the low and high aerosol loadings. If my understanding is right (that low/high aerosol loadings are relative to the median aerosol loading), then the low and high aerosol loading groups should be below and above the total observation mean, respectively (similar to Fig. 11b). Please correct me if I am mistaken and add this clarification to the text.

- Table 1: Please list the relevant PM size range measured by the Lulin Aerosol System (PM10 based on Fig 4).

- Table 3: Please add standard deviations. Please also perform a statistical significance test for each meteorological parameter between high and low aerosol loadings. Values that are statistically significant (between low vs. high loadings) can be highlighted in bold in Table 3. This would add support for the conclusions of Sect 3.5.

- Table 3: To demonstrate the semi-direct effect, RH is often mentioned in the paper; however, the instrument has diminishing sensitivity when RH approaches supersaturation (L229-231). The authors must show some evidence that the RH data for RH=[94-98%] (Table 3) is a robust enough measurement so that RH could be used to support their conclusions on the semi-direct effect. Details on the operating range from the RH sensor's manufacturer would be sufficient.

---

## Referee Comment (RC2)

Notes on 'Assessing Nonlinear Responses of Low-Level Warm Clouds Under the Impacts of Absorbing Aerosols Using the Cloud-Aerosol Mixing Ratio' by Lin et al.

This study investigates interactions between clouds and biomass burning aerosols using comprehensive in-situ measurements in a high-frequency (one-minute). They proposed a very interesting metric – cloud-aerosol mixing ratio – to separate signals from the Twomey effect and the semi-direct effect. I think they have nicely demonstrated how the evaporation due to aerosol absorption modulates the relative importance of two effects.

Overall, this is an interesting piece of work, and the proposed metric provides new insights. However, the presentation quality could be improved, and the manuscript feels somewhat longer than necessary. The writing could be more concise, and some interpretations of the key results need clarification (see my detailed comments). As such, I would recommend major revision to address the following concerns.

**Specific comments:**

The title is a bit hard to read. I'd suggest revising it to "Separating the Twomey effect and the semi-direct effect through the cloud–aerosol mixing ratio," which more clearly conveys the main points of the paper.

**Abstract:**

- The term "nonlinear" should be clearly defined, especially since it is emphasized in the title. After reading the abstract, it remains unclear whether it refers to the nonlinear response to aerosol amount or to MCr.
- Including simple equations for the key metrics (the cloud–aerosol ratio and the ACI index) would be helpful. In particular, the interpretation of the ACI index depends on whether it is defined in terms of droplet number concentration (Nd) or droplet size (<a href="https://doi.org/10.1029/2008JD011006">https://doi.org/10.1029/2008JD011006</a>).
- I think it's nice to also mention ACI index under low MCr.

L45-50: It would be good to note that such nonlinear behavior has been widely observed from space recently, and understanding it is crucial for improving climate predictions, especially in the context of continuously decreasing aerosol emissions (https://doi.org/10.1038/s41558-023-01775-5).

L62-75: This section discusses why the quantification of ACI remains so uncertain. I think it would benefit from referencing some more timely review papers that summarize

the main sources of uncertainty in the Twomey effect (e.g., <a href="https://doi.org/10.5194/acp-20-15079-2020">https://doi.org/10.5194/acp-20-7353-2022</a>).

L92: a definition of cloud water-aerosol mixing ratio should be given here

L184:  $\alpha$  refers to aerosol parameters such as AOD, PM2.5, or aerosol number concentration (Na)  $\rightarrow$   $\alpha$  refers to proxies for CCN number concentration, such as AOD, aerosol index, sulphate mass concentration, PM2.5, or aerosol number concentration (Na)... many widely used CCN proxies should be metioned.

Section 2.4: The Nd-to-aerosol susceptibility has become more commonly used than the ACI index in recent years (e.g., https://doi.org/10.1038/s41467-018-05028-4, https://doi.org/10.5194/acp-20-15079-2020). I wouldn't suggest removing this part, but perhaps linking ACINd to the Nd-to-aerosol susceptibility would be a good compromise.

Figure 3(b): what does the 'differential results' mean? Is it the derivative of curve in fig3a? Would be good to have an equation for this.

Section 2.6: SEM sounds a smart idea to remove the high-frequency signal. Could the authors explain a bit more about its physical meaning? And, the use of 60 minute is nicely justified, but I wonder to what extent the time interval can change the ACI results in section 3.

Figure 4 & Table 2: I suggest to plot the duration of each event (shown in Table 2) in Fig. 4, making it easier to read L279-284

L287-289: 'Due to significant...between high and low aerosol loads.' I don't quite see how this statement is relevant here. As I understand it, all aerosol and cloud measurements were made at the same ground level at this mountain site, so vertical co-location shouldn't be an issue in this study.

Figure 6: it's interesting to see the different ACI behaviours in low and high aerosol groups. I wonder if there is significant difference in SSA between two groups, which will help to understand the shift.

L332: why 4.5 not 4.64 - the minimal MCr with ACI index>0?

L332-333: This sentence is a bit confusing. could you clarify from what to what the proportion increases from 13% to 30%?

Figure 7: Including SSA could help explain the role of absorption in evaporation. It might be interesting to make a plot similar to Fig. 7, but with each point colored by its corresponding SSA value.

L337-338: A very confusing part again... 'Although the ACI shifted from negative to positive as aerosol concentrations increased': however, from the figure, it seems that the negative-to-positive shift occurs as MCr increases, not aerosol concentration. Also, when MCr < 2.5, the proportion of aerosol should be lower than that of cloud water, shouldn't it? Please clarify these.

**Technical corrections:**

L16: better → better

L32: 'cloud optical thickness' -> cloud optical thickness and thus cloud albedo

L40: comparing to cloud amount, LWP is even more uncertain and should be mentioned here

L57: suppress supersaturation - > reduce ambient supersaturation

L194: cloud number concentration → cloud droplet number concentration

L195: is less constrained by LWC → does not rely on the fixed-LWC assumption L233-234: Not to mention that RH measurements cannot capture supersaturation conditions: I didn't get this sentence.

---

## Author Comment (AC1)

**We appreciate the suggestion from the referee. Reviewer reports are marked in black font, our responses are marked in green bold font, and the changes to the revised manuscript are marked in blue bold font.**

**The following provides our responses and discussions regarding the major comments:**

1. MCr = 4.5 as a transition point for Twomey and semi-direct effects: Is this valid only for absorbing aerosols? I think there needs to be a control case where the ACI-MCr analysis is applied to a period characterized by scattering/non-absorbing aerosols (e.g., background, non-biomass burning (BB) air) to demonstrate that the semi-direct effect the authors observed here is strongly linked to the presence of absorbing aerosols. Since identifying the effect of absorbing aerosols is a central goal of the study, I believe a control case would be a clear demonstration that the study's findings are indeed related to the semi-direct effect.

**We acknowledge that, without additional observational evidence, the transition point of MCr = 4.5 between the Twomey and semi-direct effect can only be considered applicable to absorbing aerosols. We agree with the reviewer that it is important to further examine whether MCr remains a meaningful indicator under the influence of other aerosol types. However, during our observation period, no cloud events were captured under similar frontal conditions in the presence of non-absorbing aerosols. In real-world field campaign studies, controlling for meteorological conditions is essential when investigating aerosol-cloud interactions. Yet, similar meteorological settings often imply relatively uniform aerosol sources, making it extremely challenging to isolate the effects of different aerosol types on cloud microphysics.**

**In the case of our study site, LABS, most cloud events during spring frontal passages were associated with strongly absorbing aerosols, as demonstrated in the manuscript. To observe cloud events influenced by non-absorbing aerosols, measurements would need to extend to summer season. However, the meteorological conditions during summer are fundamentally different, with cloud formation dominated by thermal convection or deep convection. Under such conditions, meteorological dynamics exert a much stronger influence than aerosols, and the role of aerosols differs greatly**

across dynamical regimes.

Nonetheless, the fact that all cloud events captured in this study occurred under similar cold-frontal conditions provides a relatively consistent meteorological framework, allowing us to attribute the observed cloud-system variations robustly to the influence of absorbing aerosols, even without a non-absorbing aerosol control group.

Looking forward, we agree that investigating MCr across various meteorological regimes, aerosol types, and observational sites worldwide will be essential for assessing its general applicability. Such efforts may ultimately help reduce the large uncertainties associated with aerosol-cloud interactions in current climate models.

2. Regarding the possibility of "different air mass properties" (L349), the authors must demonstrate that the observed cloud responses are mainly due to the aerosol-cloud mass ratio (MCr) and not shifting air mass sources/aerosol properties over the sampling period (e.g., shift between aerosols from local vs regional, urban vs BB). As an example, Fig 6 shows the observed response of cloud microphysics grouped by PM2.5 loading. But for a causal relationship to be established, Fig 6 implies consistent aerosol source/composition/properties across different PM2.5 loadings, which the authors must demonstrate. The authors must constrain this confounding variable (aerosol composition/properties) to draw more robust conclusions about the relationship between cloud microphysics and PM2.5 loading. Alternatively, the authors can demonstrate that aerosol composition/properties are roughly consistent across PM2.5 loadings. To do this, I suggest plotting PM10 aerosol optical properties from the Lulin Aerosol System grouped by 11-D PM2.5 loading. Although this is not a one-to-one comparison, this may at least suggest that there is some degree of consistency in PM properties vs. loading.

We fully understand that maintaining consistent aerosol sources, compositions, and absorption characteristics across different aerosol concentration ranges is a crucial aspect of this study. Following the reviewer's suggestion, we analyzed the minute-resolution data from the aerosol system at LABS and plotted the results in Figure A1. As shown in Figures A1(a) and (b), the mean differences in both AAE and SAE across different aerosol concentration ranges are minimal. Most data points can be categorized as BC-type aerosols based on the optical characteristics (Schmeisser et al., 2017). It is worth noting that the standard deviation of AAE is larger in the low-concentration range (PM$_{2.5}$ < 11.1 μg m$^{-3}$), which

can be partly attributed to the increased uncertainty of the CLAP instrument under low absorption coefficients and high temporal resolution conditions (Ogren et al., 2017). For the comparison of SSA (Fig. A1(c)), the SSA values remain consistently below 0.9 regardless of aerosol concentration, indicating that the proportion of absorbing aerosols in the environment is relatively stable. In particular, the fraction of absorbing aerosols does not increase under low-concentration conditions, confirming that the consistency of aerosol optical properties does not affect the validity of the analysis in this study. Since the aerosol system at LABS is designed for long-term monitoring, we prefer to use hourly averaged data for better QA/QC reliability. Additionally, the system alternately measures $PM_{10}$ and $PM_1$ every 30 minutes, meaning that only up to 30 minutes of data are available for each size fraction per hour. This characteristic is also reflected in the sample counts of each group shown in Figure A1.

According to the reviewer's recommendation, we further analyzed the aerosol characteristics inside and outside of cloud events in Sect. 3.1 of the revised manuscript (see subsequent responses). The conclusions are consistent with the results in Figure A1. To keep the subsequent analysis more focused on the variations in cloud microphysical properties, we prefer to retain the original set of variables in Figure 6 of the manuscript.

[Figure]

**Figure A1: Aerosol optical properties categorized into 10 groups of equal sample size based on PM2.5 obtained from 11-D (the data number in each group is shown below as gray bars and values). (a) AAE, (b) SAE, (c) SSA.**

3. Washout of BB aerosol in frontal system: Won't precipitation in frontal system scavenge the biomass burning aerosols? How do the authors know that the aerosols they're sampling aren't local (e.g., Kaohsiung)? While Fig 2 show trajectories reaching the Southeast Asian Peninsula, these trajectories do not preclude the possibility of local influence and do not imply that the BB aerosols were unscavenged during the 2-3 days of transport in the warm front. The authors must either demonstrate that the source of the absorbing aerosol is Southeast Asian biomass burning or rephrase the text on the origin of the absorbing aerosol.

**We address the issue of aerosol sources and characteristics based on the following points:**

1. **Following previous studies, we focused on the peak season of biomass**

burning in the Southeast Asia Peninsula and the meteorological conditions most likely to influence Taiwan (Wang et al., 2007; Lee et al., 2011; Lin et al., 2013; Yen et al., 2013; Nguyen et al., 2020). During the observations, potential source regions and the transport of pollutants were further verified using satellite data, including satellite imagery and MODIS AOD products, as well as backward trajectories (e.g., Fig. 2(c)).

2. In addition, we analyzed vertical profiles from lidar instruments deployed by the Taiwan Ministry of Environment in the central plain region (Xitun: 24.16°N, 120.62°E; Douliu: 23.71°N, 120.54°E) before and after cloud events (Fig. S1). Using the depolarization ratio (Fig. S1(b), (d)), we identified persistent asymmetric particle signals at altitudes of 2-4 km during the observation period. This pattern differs markedly from the diurnal variations of local boundary-layer aerosols, allowing us to distinguish aerosols transported from outside Taiwan.

3. In Fig. 2(c), the backward trajectories are slightly northward relative to the Southeast Asia Peninsula. During the observation period, the cold front, influenced by high-pressure systems, extended mainly from northeast to southwest and moved eastward. Under these meteorological conditions, biomass-burning aerosols could be transported by the pre-frontal westerlies (Huang et al., 2020) and couple with the frontal cloud system. Since the frontal generation region is primarily located north of the main biomass-burning areas, the backward trajectories are considered reasonable.

4. Following the reviewer's suggestion, MODIS fire hotspot data from March 1-12, 2024, were added to Fig. 2(c). The figure confirms that the primary fire activity is indeed located over the Southeast Asia Peninsula.

5. We agree that aerosols can be affected by wet removal processes during transport. However, wet deposition generally removes aerosols from the ambient environment as a whole. Light-absorbing aerosols, such as black carbon, are more hydrophobic and thus less sensitive to wet removal (McMeeking et al., 2011; Ohata et al., 2016; Pöhlker et al., 2023). As shown in Fig. A1(c), the single scattering albedo indicates that the proportion of light-absorbing aerosols remains high even at low aerosol concentrations, and minimal changes in SSA before and after cloud

events (Fig. 4(d), see subsequent analysis) suggest that light-absorbing aerosols persist in cloud systems despite potential wet removal effects.

6. We also agree that the description and attribution of biomass-burning or absorbing aerosols should be precise. In this study, the observed optical properties of aerosols provide direct evidence, and we primarily refer to absorbing aerosols in the Results and Discussion sections. Based on the combined analyses, we are confident that biomass-burning aerosols from the Southeast Asia Peninsula constitute a major source of the observed light-absorbing aerosols during the study period.

L170: …with frontal systems. "In addition, this study employed the NOAA HYSPLIT model to generate 3-day backward trajectories, MODIS fire hotspot data from Terra and Aqua (Fig. 2 (c)), and lidar observations deployed by the Taiwan Ministry of Environment in central Taiwan (Fig. S1) as supporting evidence to assess the potential impact of long-range transport of biomass-burning aerosols. The results indicate that, during the cloud events, air masses surrounding LABS may have traversed affected regions, such as the southern coastal areas of China and the Southeast Asian Peninsula, while the fire hotspot data further highlight the Southeast Asian Peninsula as potential major sources of biomass burning during this period."

[Figure]

**Figure S1: Lidar observations at the Xitun site (24.16°N, 120.62°E) and Douliu site (23.71°N, 120.54°E) of the Taiwan Ministry of Environment from 29 February to 12 March 2024. Panels (a) and (c) represent the NRB signal intensity, while panels (b) and (d) show the depolarization ratio.**

[Figure]

**Figure 2: (a) Himawari 8 true color satellite image and (b) surface chart from Central Weather Administration (CWA) at 08 LT (UTC +8), March 7, 2024. (c) Fire points data from March 1 to March 12, 2024, were plotted together with the timing of major cloud events (DD_HH, LT) using 72-hour backward trajectories. Fire hotpot data were obtained from MODIS onboard Terra and Aqua, considering only those with confidence levels above 80%. The backward trajectories were calculated using the NOAA HYSPLIT model (GFS 0.25° × 0.25°) and displayed at 6-hour intervals.**

4. I strongly recommend adding a conceptual aerosol-cloud figure to visually explain the key findings of the paper, which can be used as a graphical abstract as well.

We thank the reviewer for the suggestion. Our schematic diagram (Fig. 13) will be added to the Conclusion section to provide an integrated summary, thereby clarifying the main points of the manuscript.

L488: "The overall microphysical variations in aerosol-cloud interactions can be divided into two dominant regimes, the semi-direct effect and the Twomey effect, whose relationships with MCr are illustrated in Figure 13. A low MCr value indicates that each unit of aerosol within the cloud system is associated with a relatively limited amount of liquid water. Under this condition, the entrainment of additional absorbing aerosols is mainly governed by the semi-direct effect, resulting in the dissipation of small cloud droplets (Fig. 13). In contrast, a high MCr value represents an environment with abundant liquid water relative to aerosol loading, where the increase in aerosols enhances the formation of numerous small droplets through microphysical processes described by the Twomey effect (Twomey, 1974, 1977)." Notably…

[Figure]

Figure 13: A schematic illustration showing how aerosol entrainment leads to different cloud microphysical responses under varying MCr conditions.

The following provides our responses and discussions regarding the minor comments:

- L11-12/L81-86/L106-108: The trajectories from Fig. 2c appear to mainly pass through southeast China and the surrounding waters around Taiwan. The trajectories that do reach the Southeast Asian Peninsula are quite few (only two). The specific source of the absorbing aerosol should not affect the conclusions of the paper, but since Southeast Asian biomass burning was mentioned several times, more evidence that the aerosols originated from biomass burning in that region rather than southeast China is required. For example, the authors could add hotspots over the period of the backtrajectories in Fig. 2c to reinforce claim over the source of absorbing aerosols.

**The detailed response is provided in major comment 3.**

- L80-81: The sentence "Observational studies have…" should be moved to the paragraph before this one for better organization.

**L49: … on cloud droplet size and number (Feingold et al., 2001; Saponaro et al., 2017; Chen et al., 2021; Jia and Quaas, 2023). "Observational studies have also shown that the water content in the atmosphere plays a critical role in determining whether aerosols ultimately enhance or suppress cloud formation (Zhang and Zuidema, 2019, 2021)."**

- L102: Please mention if the aerosol measurements at LABS have a drying tube for in-cloud periods. Please also specify if the optical properties in Fig 4 refer to dried or ambient aerosols.

**The aerosols measured by the LABS aerosol system were all dry aerosols. For both in-cloud and out-of-cloud sampling, the relative humidity inside the sampling line remained below 40%. Therefore, the data presented in Figure 4 represent observations of dry aerosols. After connecting the dryer to the 11-D inlet, aerosol sampling and analysis were also conducted under conditions with relative humidity below 40%. A corresponding clarification has been added to Section 2.2, where the instrumentation is described.**

**L149: … comprehensive analysis. "The aerosol system was equipped with a heated drying unit at the inlet, maintaining the relative humidity of the sampled air below 40%. Therefore, even during cloud events, the measurements represent the properties of dry aerosols. Similarly, the 11-D system also used a drying tube at the inlet (Fig. 1 (c)) to keep the relative humidity of the sampled air below**

**40%.”**

- L129: Please specify if the 11-D aerosol measurement during cloud events refers to interstitial aerosol and/or the in-droplet aerosol.

**In this study, both the 11-D and the aerosol system sampled dry aerosols or aerosols dried prior to measurement. No additional wet/dry filtering devices were installed upstream of the sampling line, and neither instrument performed dedicated comparative measurements between dry and wet aerosols. Consequently, the aerosols measured in this study include both interstitial aerosols and in-droplet aerosols.**

- L237: Please justify the use of PM2.5 over PM10 in the formula for MCr.

**We consider PM$_{2.5}$ to be an important global indicator for assessing aerosol loading. Moreover, PM$_{2.5}$ measurements (from stations equipped with high-precision instruments to those using low-cost sensors) are now widely applied, facilitating future scientific investigations of aerosol-cloud interactions worldwide, as well as multi-site observational studies targeting the effects of absorbing aerosols on horizontally developed clouds. In addition, observations during our experimental period show that the temporal variations of PM$_{2.5}$ and PM$_{10}$ exhibit similar trends (Fig. S2 (d)). Therefore, PM$_{2.5}$ was selected as the basis for our calculations.**

- L238: Please clarify what aerosol size range N_a refers to.

**The description of the N$_a$ range has been added in Lines 230-231.**

**L229: … 11-D measurements (only aerosols within the instrument's measurement size range (0.253-35.15 μm) were included in the particle number calculation), the study primarily focuses on MCr for quantitative analysis. …**

- Sect 3.1: It would be helpful to the reader if the authors explicitly stated in this section that the aerosol optical properties refer to PM10 while the rest of the paper refers to PM2.5.

**We thank the reviewer for the suggestion. The revisions to the manuscript are detailed below.**

- L260/Sect 3.1: The purpose of Sect 3.1 (PM10 optical properties) needs to be explicitly/more clearly explained in the context of the rest of the paper which uses PM2.5 and focuses only on cloud events. I suggest reframing Sect 3.1 to introduce the cloud events (Table 2) and then describe PM10 aerosol properties before, during, and after each cloud event (Fig 4). This would then flow more smoothly into Sect 3.2. If the LABS optical data is valid in-cloud (e.g., if a dryer was placed upstream of the instruments), it would be informative to include this information as additional columns in Table 2 and added to the discussion in this section.

**In the reorganized manuscript, we place greater emphasis on the analysis of individual cloud events, and the averaging procedure in Fig. 4 is now based on these cloud events, as detailed below. Descriptions related to aerosol sampling by the aerosol system are provided in Section 2.2. Table 2 has been updated to include the average PM$_{2.5}$ and LWC for each cloud event for subsequent analysis.**

- L260/Sect 3.1: I recommend adding a meteorological time series to add context to Fig 4 and Sect 3.1 (with similar markings/shadings to show when cloud events occurred). This would help establish a stronger link between the key findings and the physical environment during the sampling period. This meteorological time series could be in the supplementary information.

**Figure S2 and the manuscript are provided below. In Figure S2(c), it can be seen that not all periods with low visibility and relative humidity near 100% were classified as continuous cloud events. This is because some of these periods were caused solely by precipitation, and sometimes the CDP laser windows were wet or contaminated by rain, resulting in poor-quality observations that were excluded prior to analysis.**

- Fig 4: Please mark each cloud event from Table 2 (e.g., grey areas spanning the start until end of each event). Also mention in caption that daily averages and STDs are also shown.

**In Figure 4, the calculation of averages and standard deviations has been revised to reflect values within individual cloud events, allowing a more focused comparison of whether aerosol optical properties are consistent across different cloud events.**

**L261:**

Figure 4 presents the aerosol observations from the LABS aerosol system, with the corresponding meteorological information shown in Fig. S2. In this section, the aerosol optical properties of aerosols are analyzed using PM10 measurements from the aerosol system, whereas PM2.5 observations from the 11-D instrument are primarily used in the other sections. The gray-shaded intervals denote continuous cloud events lasting more than three hours and detected by CDP (Table 2). Based on the absorption and scattering coefficients as well as aerosol mass concentrations (Fig. 4 (a), (b) and Fig. S2 (d)), it is evident that during cloud events accompanied by significant precipitation (Fig. S2 (d)), aerosol concentrations decreased due to washout effect. However, in periods without precipitation signals prior to 10 March, aerosol concentrations clearly increased, indicating continuous long-range transport of aerosols to the LABS region. Overall, higher pollutant levels were observed during the early stage of the cloud event on March 1 (Event 1), both the early and late stages of the event on March 7 (Event 2), and throughout the event on March 9 (Event 3). In contrast, lower concentrations occurred during the precipitation periods on March 1 and March 7, and during the cloud events on March 10 and 11 (Event 4 and Event 5).

Further analysis was conducted using AAE and SAE to evaluate the light absorption and scattering abilities of aerosols at blue and red wavelengths (Fig. 4 (c), (d)). Except for the Event 1, when the mean AAE value was significantly lower than 1 ($0.61 \pm 0.12$), the average AAE during other cloud events was close to 1, suggesting minimal wavelength dependence of aerosol absorption. The SAE remained relatively stable throughout the observation period but showed a slight decrease on March 9 ($1.47 \pm 0.01$), possibly indicating the presence of coarse-mode aerosols related to local sources. The single-scattering albedo (SSA), used to evaluate the overall aerosol absorption capacity (Fig. 4(e)), was generally below 0.9 during most of the observation period except on March 12. Notably, no significant correlation was found between SSA values and aerosol concentrations before and after cloud events, suggesting that despite effects of wet deposition, the proportion of light-absorbing aerosols within the total aerosol population remained relatively consistent. In some cloud events (Events 2 and 5), SSA even decreased during precipitation (Fig. S2 (d)), indicating an increased fraction of absorbing aerosols, likely due to the relatively hydrophobic nature of black carbon (McMeeking et al., 2011; Ohata et al., 2016; Pöhlker et al., 2023), which reduces its susceptibility to wet removal compared to more hygroscopic

components.

The overall…

Table 2: the start, end local times and the mean of PM$_{2.5}$, LWC of continuous cloud events (UTC +8).

| Cloud Event | Start Time (LT) | End Time (LT) | Average PM$_{2.5}$ | Average LWC |
|---|---|---|---|---|
| Event 1 | 3/1 09:37 | 3/1 15:40 | 7.8 ± 3.1 | 0.130 ± 0.130 |
| Event 2 | 3/7 09:16 | 3/8 07:28 | 11.4 ± 9.3 | 0.138 ± 0.126 |
| Event 3 | 3/9 11:07 | 3/9 15:27 | 20.3 ± 4.5 | 0.163 ± 0.164 |
| Event 4 | 3/10 02:13 | 3/10 06:00 | 2.3 ± 1.6 | 0.229 ± 0.085 |
| Event 5 | 3/11 09:47 | 3/12 05:21 | 2.5 ± 3.2 | 0.147 ± 0.121 |

[Figure]

Figure 4: PM$_{10}$ aerosol optical properties observed by the aerosol system at LABS, the gray-shaded intervals represent continuous cloud events. (a) absorption coefficient; (b) scattering coefficient; (c) Absorption Ångström Exponent (AAE, red) and Scattering Ångström Exponent (SAE, orange); (d)

**Single Scattering Albedo (SSA), and gray dash line means SSA = 0.9.**

[Figure]

**Figure S2: Hourly averages of meteorological parameters and aerosol mass concentrations observed at LABS from March 1 to 12, 2024: (a) wind speed and direction; (b) temperature (green) and pressure (red); (c) visibility (orange) and relative humidity (dark blue); (d) PM$_{10}$ (dark red) and PM$_{2.5}$ (red) measured by the 11-D instrument, along with hourly precipitation (light blue bars).**

- L285/Section 3.2: This section is discussed in terms of Group Number. To keep the discussion physically tied to the atmosphere, I suggest that the discussion be based on the PM2.5 concentration ranges (physically based) rather than group number (statistically based). Please rephrase this section so that Group Numbers are replaced by PM2.5 concentration ranges. This will also make the section more succinct. Please apply this change to other sections that mention the PM

groups too (e.g., L463).

- L296: Please define ED here or sooner if it's mentioned.
- Fig 6: please mention in the caption how many samples are in each group.
- Fig 6: Because of the uneven width of each group (a consequence of the equal-number groupings), I suggest the x-axis should be PM2.5 concentration rather than group number. This will show the true scale of the x-axis to make it clear that the observed trends are most conclusive for PM2.5 < 10 micrograms m-3 (where most data are). For clarity, one variable can take up one panel (so 6 panels).
- Fig 6: Instead of infinity, Group 10 should list the max PM2.5 concentration recorded during the period.

**The revisions to the manuscript and the figure are detailed below.**

**L292: Figure 6 shows the aerosol loading in the environment, ranging from low to high. "The maximum mean value of $N_d$ (370 cm$^{-3}$) was observed when the PM$_{2.5}$ concentration ranged between 1.4 and 2.6 μg m$^{-3}$ (Fig. 6 (a)), whereas LWC reached the maximum mean value (0.219 g m$^{-3}$) within the lower PM2$_{2.5}$ concentration range of 0.4-1.4 μg m$^{-3}$(Fig. 6 (c)). Notably, when PM$_{2.5}$ ≥ 2.6 μg m$^{-3}$, both $N_d$ and LWC exhibited a decreasing trend as environmental aerosol concentrations increased. The variation in effective diameter (ED) can be characterized by a threshold at PM$_{2.5}$ = 5.7 μg m$^{-3}$ (Fig. 6 (b)). When PM$_{2.5}$ concentrations were below this value, ED decreased with increasing aerosol loading. However, when PM$_{2.5}$ ≥ 5.7 μg m$^{-3}$, ED maintained an average size or slightly increased despite the higher aerosol concentration. A further comparison of weather data across different PM$_{2.5}$ concentration ranges revealed that when PM$_{2.5}$ ≥ 21.7 μg m$^{-3}$, the average temperature (4.3°C, Fig. 6 (d)) and specific humidity (7.0 g kg$^{-1}$, Fig. 6 (f)) reached the lowest values, possibly constrained by specific time and weather conditions. Additionally, relative humidity (Fig. 6 (e)) displayed a decreasing trend with increasing environmental aerosol loading, and the standard deviation of relative humidity increased, reaching the lowest value (93.4%) and the highest standard deviation (7.1%) between PM$_{2.5}$ concentrations of 16.4 and 21.7 μg m$^{-3}$."**

**L305: … below 4.2 μg m$^{-3}$. "Compared with environments where PM$_{2.5}$ ≥ 5.7 μg m$^{-3}$, those with extremely low aerosol concentrations (PM$_{2.5}$ < 2.6 μg m$^{-3}$) exhibited noticeably higher LWC, highlighting the substantial differences in cloud liquid water among different aerosol loading conditions. The range of**

PM$_{2.5}$ between 2.6 and 5.7 μg m$^{-3}$ appeared to represent a transitional phase. In environments with relatively low average aerosol concentrations, higher LWC coexisted with higher N$_d$ and lower ED values, indicating that sufficient water availability allowed more aerosol particles to be activated." It is important to note that … In the high aerosol loading environment "(PM$_{2.5}$ ≥ 5.7 μg m$^{-3}$),)," increased aerosol concentrations result in a decrease in N$_d$ and a slight increase in ED …

L460: "In a comparative analysis involving 10 data groups stratified by both PM$_{2.5}$ concentration levels and sample size considerations, a distinct trend emerged: in the low-concentration (PM$_{2.5}$ < 2.6 μg m$^{-3}$), an increase in PM$_{2.5}$ was associated with an increase in N$_d$ and a significant decrease in ED. However, in the high-concentration when PM$_{2.5}$ ≥ 5.7 μg m$^{-3}$, further increases in PM$_{2.5}$ corresponded to a decrease in N$_d$ and a slight increase in ED. These results suggest a transition in cloud microphysical responses under elevated aerosol loading conditions." …

[Figure]

**Figure 6: Cloud microphysical and weather data categorized into 10 groups of equal sample size based on PM$_{2.5}$ obtained from 11-D (the data number in each group is shown below as gray bars and values). (a) N$_d$, (b) ED, (c) LWC from CDP observation results; (d) temperature, (e) relative humidity and (f) specific humidity are weather results measured at LABS.**

- L318/Sect 3.3: This section requires major revision for clarity, specifically more explicitly connecting the conclusions in the text to the figures so it will be easier for the reader to understand how the data supports the authors' conclusions. See my comments below.
- L336-340: it is not clear in any figure that the PM2.5 > LWC when MCr < 2.5 (since MCr is a ratio, a drop in MCr could be due to changes in LWC and/or PM2.5 and MCr < 2.5 does not necessarily imply that PM2.5 > LWC). It is also not evident from Fig 7 that ACI shifts from negative to positive as aerosol concentrations increase, please provide some x/y values to orient the reader to which part of Fig 7 to refer to. The panels in Fig. 7 are not clearly referenced in the text (i.e., only Fig. 7 is mentioned so it is unclear which panel to refer to while reading the text).
- L344-345: It is not evident from the figures that Event 3 occurred under high aerosol concentrations and low LWC. To add physical context to each cloud event, please add new columns to Table 2 with statistics on cloud microphysical and aerosol properties (e.g., mean+STD PM2.5) for each event. This would help with contextualizing the differences discussed in Sect 3.3 and explaining differences in how each event fills the 2D space in Fig. 7.

We agree that the original description was not sufficiently precise. In this manuscript, MCr refers to a metric that simultaneously accounts for aerosol loading and liquid water content within the cloud system. Physically, MCr represents the amount of liquid water in the environment associated with each unit of aerosol concentration; therefore, it is not simply a direct comparison between PM$_{2.5}$ and LWC as implied in the earlier text.

In Figure 7, each MCr point reflects the hourly averaged ratio between PM$_{2.5}$ and LWC, whereas ACI quantifies the perturbations of aerosols and cloud droplets within each hour. Because the use of a moving-average approach produces a high degree of temporal continuity, successive MCr-ACI points are strongly correlated. To highlight the key features without making the figure overly cluttered, the full dataset is shown using grayscale density shading, while

individual data points plotted at 30-minute intervals illustrate the distribution across different cloud events.

For the case on March 7 (Event 2; Lines 336-340), neither the density shading nor the 30-minute sampling captures the behaviour described in the text; this feature only appears in the computed values and represents an extremely small number of outliers. Since this rare behaviour does not materially affect the subsequent analyses, we have shortened and reorganized the description in Section 3.3.

Following the reviewer's suggestion, Table 2 has been updated to include the mean and standard deviation of $PM_{2.5}$ and LWC for each cloud event. In the latter part of Section 3.3, we focus on how differences in $PM_{2.5}$, LWC, and MCr among cloud events manifest in variations in ACI, thereby underscoring the importance of MCr. To further sharpen the analysis, Figure 7 now presents only the MCr results, whereas the NCr results have been moved to the Supplement (Fig. S3). The complete revisions are detailed below.

L319-331:
L323: … display a positive ACI index value. "Of these, over 95% (334 out of 351) occurred in regions where MCr exceeded 4.5 (with the minimum MCr value within the reasonable range for ACI index > 0 being 4.64), especially when positive ACI values within the reasonable range (0-0.33, Fig. 7). This finding suggests that under the influence of high aerosol concentrations, cloud systems tend to adjust by reducing the cloud droplet number concentrations most of the time. As shown in Figure S3, the distribution of NCr-ACI reveals a similar tendency to that observed in the MCr-ACI relationship. After constraining aerosol properties and meteorological conditions, negative ACI index can be meaningfully interpreted (as described in Sect. 2.4). However, most of the previous studies have focused only on positive region. It is important to note that negative values do not have a defined "reasonable range", they simply indicate that an increase in aerosol concentration is accompanied by a decrease in cloud droplets number." The physical mechanism …

L332-354:
Comparative analysis among individual cloud events reveals several notable trends (Fig. 7). All events show a consistent relationship between MCr and ACI, where smaller ACI values (more negative) correspond to lower liquid water content per unit aerosol abundance (i.e., smaller MCr). A closer examination of cases with MCr > 4.5 shows that 31% of ACI values (135 data points) are positive

in Event 2, and 93% (148 data points) are positive in Event 4. In contrast, although ACI values in Events 1 and 5 tend to increase with MCr, most of them remain negative. These findings suggest that short-term variations in cloud systems are more sensitive to environmental changes induced by absorbing aerosols, where increased aerosol loading can lead to the dissipation of cloud droplets. It is noteworthy that although Event 3 exhibited relatively high mean LWC (the second highest among the five consecutive cloud events), it occurred under high aerosol loading conditions (Table 2). Consequently, the liquid water content per unit aerosol abundance was relatively low (MCr ≈ 4; Fig. 7), and the calculated ACI values remained negative throughout the event (Fig. 7). This result suggests that aerosol-cloud interactions under the influence of absorbing aerosols cannot be interpreted solely based on aerosol concentration. In particular, since the liquid water content varies substantially among real world cloud events, MCr provides a more representative measure of the actual aerosol loading within clouds.

[Figure]

Figure 7: Scatter and density plots of the MCr-ACI index for the continuous cloud events. The density plot illustrates the overall distribution of all calculated results, while the scatter plot shows the distribution of individual cloud events using different colors and symbols (one point shown for every 30 calculations).

[Figure]

**Figure S3: Scatter and density plots of the NCr-ACI index for the continuous cloud events. The settings are identical to those used in Figure 7.**

- L357: "sufficient liquid water content" please clarify: sufficient for what? For the MCr ratio to be applicable? Or for the Twomey effect to be dominant?
- L358: "division" please clarify what this division is for. If this division/threshold of MCr refers to the point where Twomey or semi-direct effects dominate, please specify this explicitly to make the purpose of Sect 3.4 clearer.

We have provided a more detailed description regarding this point.

L356: … mixing ratio and ACI index. "To further clarify this relationship, different MCr thresholds were applied as screening criteria (Fig. 8) to identify suitable divisions that distinguish conditions under which aerosols in cloud systems exist in liquid-water-sufficient environments (where the Twomey effect tends to dominate) and those in liquid-water-limited environments (where the semi-direct effect prevails)." As shown in Fig. 8, …

- L375-383: Please state the implication/importance behind why ED does not strongly correlate with ln(PM2.5). Please also justify the use of ln(PM2.5) rather than PM2.5.

The difference between ED and $N_d$ in the calculation of the ACI index can primarily be attributed to the following two factors:

1. Intrinsic differences between in-situ instruments and satellite retrievals when deriving cloud microphysical parameters.

    Although satellites also provide $N_d$ and $r_e$, the retrieved $N_d$ and $r_e$ can in practice be expressed as functions of LWP and COT (see L175-199 of the manuscript). Therefore, in theory, $ACI_{Nd}$ can be equivalent to $ACI_{re}$ when LWP is constrained; however, this mathematical dependence is also one of the major sources of uncertainty in satellite-based ACI estimates.

    In contrast, in-situ cloud droplet probes derive $N_d$ and ED (ED $= 2r_e$) directly from the fundamental measurements of droplet counts and sizes. Thus, $N_d$ and ED may be regarded as $N_d$ (count) and ED (count, size). Unless the constraints are applied specifically to droplet counts or droplet sizes, differences between $ACI_{Nd}$ and $ACI_{re}$ are expected. But such constraints would not be physically meaningful in this context.

2. A substantial part of the $ACI_{re}$ uncertainty arises from the choice of the LWC or LWP interval width.

    In principle, a narrower interval yields a more reliable estimate. However, achieving extremely narrow intervals is challenging for real-world observations (Quaas et al., 2020). Considering sample size limitations in this study, we used an interval of $\Delta$LWC $= 0.1$ g m$^{-3}$, which may indeed introduce additional uncertainty.

In Figure 8, the plotted relationship is between $\ln(PM_{2.5})$ and $\ln(r_e)$, corresponding to the two parameters used to compute $ACI_{re}$ (Eq. (5); see updated Figure 8). Since ED $= 2r_e$, the y-axis was converted from $\ln(r_e)$ to ED to provide a more intuitive representation of the ED distribution during our observation period.

- L386-388: Can this conclusion about the droplet size distribution be seen in Fig. 11-12? If so, please mention that to support this conclusion.

Figure A2 compares small droplets (<10 μm, blue boxes) and large droplets (>12 μm, orange boxes) under high and low aerosol loading. Under high aerosol loading, the number of small droplets is noticeably reduced, whereas more droplets appear in the larger-size range. This pattern remains consistent regardless of whether LWC is constrained. Table A1 further shows that high

aerosol loading is associated with fewer cloud droplets ($N_d$) but a larger effective diameter (ED), which is consistent with the discussion in L386-388.Additional details regarding the discussion related to Fig. 11(b) and Fig. 12(b) are provided in Lines 429–445 of the manuscript.

Table A1: Mean $N_d$ and ED across different conditions (bold values indicate statistically significant differences between the two groups according to an independent t-test (P-value < 0.05)).

|  |  | $N_d$ (cm$^{-3}$) | ED (μm) |
|---|---|---|---|
| MCr [3, 4.5] | High aerosol loading | 217 ± 150 | **11.14 ± 2.34** |
|  | Low aerosol loading | 224 ± 158 | **9.51 ± 2.03** |
| MCr [3, 4.5] in LWC [0.1, 0.2] | High aerosol loading | **267 ± 135** | **11.71 ± 1.91** |
|  | Low aerosol loading | **360 ± 164** | **10.53 ± 1.72** |

[Figure]

Figure A2: (a) Figure 11 (b); (b) Figure 12 (b).

- L454/Sect 4: Please describe some suggestions for future analysis based on the main conclusions of the manuscript.

L492: …concentrations and clouds. "Fundamentally, the cloud-aerosol mixing ratio framework moves beyond considering the influence of aerosol emissions alone and instead incorporates the coupled state between aerosols and the environment or between aerosols and the cloud system. This provides a more comprehensive perspective for understanding the feedback mechanisms involved in aerosol-cloud interactions. Although this study, through prior screening and constraining of meteorological and aerosol conditions, ultimately converges on

the role of absorbing aerosols, the impacts of other aerosol types and their actual in-cloud loadings at different locations around the world remain unresolved. Nevertheless, given the widespread availability of global PM2.5 observations, future work will be able to leverage extensive monitoring networks in combination with in-situ cloud-water measurements to further investigate these processes."

- Fig 5: please mention the PM size range for these measurements in the caption.

[Figure]

**Figure 5: Classification of PM$_{10}$ aerosols at LABS from March 1 to March 12, 2024 (Classification method adapted from Schmeisser et al. (2017)), BC means black carbon type, BrC means brown carbon type.**

- Fig 8: Please include a panel to show the number of data points included in the calculation of the y-axis per threshold value (x-axis) to demonstrate that the converging trends mentioned are robust.

We provide the number of data points in the Supplementary Information (Supplementary Figure 4). Because Figure 8 classifies all available observations into different MC$_r$ intervals, the sample sizes across intervals naturally vary. Nevertheless, in MCr threshold = 3.7-4.9, both groups contain near or more than 1000 data points, which is within a reasonable range for statistical analysis.

[Figure]

**Figure S4: Number of data points within different MCr threshold intervals in Figure 8.**

- Fig 8b: Please explicitly state in the caption what two quantities the correlation is being calculated for (e.g., ACI and MCr for data where MCr > threshold?). Please apply this to Fig. 9 too perhaps by changing the notation of r(ACI) (e.g., r(ACI, MCr)) for clarity.
- Fig 9: For visual clarity, please use different colors (rather than shades of blue) for different LWC groupings.

**The updated figure is provided below for reference.**

[Figure]

**Figure 8: (a) ACI values, (b) correlation coefficients, and (c) RMSE are calculated based on different MCr thresholds. Blue markers represent data subsets with MCr values above the given threshold, while red markers indicate those below it. Circles correspond to calculations using PM2.5 as α, and triangles denote those using N_a as α.**

[Figure]

**Figure 9: Scatter plot of ln(PM₂.₅) and ED, along with $\mathrm{ACI}_{r_e}$ calculation results under specified LWC conditions during the observation period. (a) Results for all observed data; (b) MCr [4.5, 6]; (c) MCr [3, 4.5].**

- Fig 11: Please double check the coloring of the aerosol size distributions in (a). The observation mean size distribution is not between the low and high aerosol loadings. If my understanding is right (that low/high aerosol loadings are relative to the median aerosol loading), then the low and high aerosol loading groups should be below and above the total observation mean, respectively (similar to Fig. 11b). Please correct me if I am mistaken and add this clarification to the text.

In Figure 11, the black line represents the mean values calculated from all observations during the study period, rather than being constrained within specific MCr intervals. Therefore, the black lines in panels (a) and (b) are identical. This design highlights the contrast between the MCr-conditioned results and the overall observational mean. It further demonstrates that if aerosol loading were classified solely based on PM$_{2.5}$, the differences shown in Figure 11(a) would be obscured because certain cases would be misclassified as "clean events," thus masking the actual signal. In Figure 12, the black line represents the mean values of all data with LWC between 0.1 and 0.2 g m$^{-3}$ during the study period. The updated figure labels are shown below.

Figure 11: Size distribution of aerosols (triangles) and cloud droplets (circles). The black line represents the mean of all observations during the study period (not constrained by MCr intervals). Red and green lines follow the same conditions as in Figure 10. (a) Red and green lines limited to the MCr range [4.5, 6]; (b) Red and green lines limited to the MCr range [3, 4.5].

Figure 12: Same as Fig. 11, but all data are limited to the LWC range of 0.1-0.2 g m$^{-3}$, including the black line, which represents the mean of all observation data within this LWC range.

- Table 1: Please list the relevant PM size range measured by the Lulin Aerosol System (PM10 based on Fig 4).

Table 1. Parameters Used in This Study.

| Manufacturer/ Instrument | Parameters | Acronym | Unit | Time Resolution |
|---|---|---|---|---|
| | MetOne, T200 | Temperature | T | °C | |
| | MetOne, 092 | Pressure | P | hPa | |
| Lulin Weather System | MetOne, 083D | Relative Humidity | RH | -- | 1 min |
| | MetOne, 034B | Wind Speed | WS | m s$^{-1}$ | |
| | | Wind Direction | WD | degree | |

| | | | | | |
|---|---|---|---|---|---|
| | VAISALA, PWD22 | Visibility | -- | km | |
| | MetOne, 370 | Rainfall Intensity | -- | mm hr$^{-1}$ | 1 hour |
| **Lulin Aerosol System (PM$_{10}$ data was used in this study.)** | GMD, CLAP | Absorption Coefficient | -- | Mm$^{-1}$ | |
| | TSI 3563, Nephelometer | Scattering Coefficient | -- | Mm$^{-1}$ | |
| | -- | Absorption Ångström Exponent | AAE | -- | 1 hour |
| | -- | Scattering Ångström Exponent | SAE | -- | |
| | -- | Single Scattering Albedo | SSA | -- | |
| Aerosol-Cloud Microphysics Monitoring System | DMT, CDP | Cloud droplet number concentration | N$_d$ | cm$^{-3}$ | 1 min |
| | | Cloud Number Size Distribution | CSD | cm$^{-3}$ | |
| | | Effective Diameter | ED | μm | |
| | | Liquid Water Content | LWC | g m$^{-3}$ | |
| | GRIMM, 11-D | Aerosol Number Concentration | N$_a$ | cm$^{-3}$ | 1 min |
| | | Aerosol Number Size Distribution | ASD | cm$^{-3}$ | |
| | | PM$_{2.5}$ | -- | μg m$^{-3}$ | |

- Table 3: Please add standard deviations. Please also perform a statistical significance test for each meteorological parameter between high and low aerosol loadings. Values that are statistically significant (between low vs. high loadings) can be highlighted in bold in Table 3. This would add support for the conclusions of Sect 3.5.
- Table 3: To demonstrate the semi-direct effect, RH is often mentioned in the paper; however, the instrument has diminishing sensitivity when RH approaches supersaturation (L229-231). The authors must show some evidence that the RH data for RH=[94-98%] (Table 3) is a robust enough measurement so that RH could be used to support their conclusions on the semi-direct effect. Details on the operating range from the RH sensor's manufacturer would be sufficient.

The MetOne 083D RH sensor operates over a relative humidity range of 0-100%, with an accuracy of ±3% in the 90-100% range. To improve transparency regarding the instruments used in this study, we have added the corresponding manufacturer and model information to Table 1 so that readers may easily reference and verify the specifications.

Table 3: Mean temperature and humidity across different conditions (bold values indicate statistically significant differences between the two groups according to an independent t-test (P-value < 0.05)).

| | | Temperature (°C) | RH (%) | Specific Humidity (g kg$^{-1}$) |
|---|---|---|---|---|
| **MCr [4.5, 6]** | High aerosol loading | 5.8 ± 1.3 | **98 ± 2** | 7.85 ± 0.73 |
| | Low aerosol loading | 5.6 ± 1.0 | **98 ± 1** | 7.76 ± 0.53 |
| **MCr [4.5, 6] in LWC [0.1, 0.2]** | High aerosol loading | 5.8 ± 1.2 | **98 ± 1** | 7.83 ± 0.68 |
| | Low aerosol loading | 5.8 ± 0.9 | **98 ± 1** | 7.81 ± 0.45 |
| **MCr [3, 4.5]** | High aerosol loading | **6.4 ± 1.1** | 95 ± 6 | **7.92 ± 0.71** |
| | Low aerosol loading | **5.5 ± 1.0** | 97 ± 2 | **7.62 ± 0.52** |
| **MCr [3, 4.5] in LWC [0.1, 0.2]** | High aerosol loading | **6.8 ± 0.9** | 94 ± 6 | **8.02 ± 0.69** |
| | Low aerosol loading | **5.8 ± 1.1** | 98 ± 2 | **7.80 ± 0.56** |

**Reference**

Huang, H.-Y., Wang, S.-H., Huang, W.-X., Lin, N.-H., Chuang, M.-T., da Silva, A. M., and Peng, C.-M.: Influence of Synoptic-Dynamic Meteorology on the

Long-Range Transport of Indochina Biomass Burning Aerosols, *Journal of Geophysical Research: Atmospheres*, 125, e2019JD031260, https://doi.org/10.1029/2019JD031260, 2020.

Lee, C.-T., Chuang, M.-T., Lin, N.-H., Wang, J.-L., Sheu, G.-R., Chang, S.-C., Wang, S.-H., Huang, H., Chen, H.-W., Liu, Y.-L., Weng, G.-H., Lai, H.-Y., and Hsu, S.-P.: The enhancement of PM2.5 mass and water-soluble ions of biosmoke transported from Southeast Asia over the Mountain Lulin site in Taiwan, *Atmospheric Environment*, 45, 5784-5794, https://doi.org/10.1016/j.atmosenv.2011.07.020, 2011.

Lin, N.-H., Tsay, S.-C., Maring, H. B., Yen, M.-C., Sheu, G.-R., Wang, S.-H., Chi, K. H., Chuang, M.-T., Ou-Yang, C.-F., Fu, J. S., Reid, J. S., Lee, C.-T., Wang, L.-C., Wang, J.-L., Hsu, C. N., Sayer, A. M., Holben, B. N., Chu, Y.-C., Nguyen, X. A., Sopajaree, K., Chen, S.-J., Cheng, M.-T., Tsuang, B.-J., Tsai, C.-J., Peng, C.-M., Schnell, R. C., Conway, T., Chang, C.-T., Lin, K.-S., Tsai, Y. I., Lee, W.-J., Chang, S.-C., Liu, J.-J., Chiang, W.-L., Huang, S.-J., Lin, T.-H., and Liu, G.-R.: An overview of regional experiments on biomass burning aerosols and related pollutants in Southeast Asia: From BASE-ASIA and the Dongsha Experiment to 7-SEAS, *Atmospheric Environment*, 78, 1-19, https://doi.org/10.1016/j.atmosenv.2013.04.066, 2013.

McMeeking, G. R., Good, N., Petters, M. D., McFiggans, G., and Coe, H.: Influences on the fraction of hydrophobic and hydrophilic black carbon in the atmosphere, *Atmos. Chem. Phys.*, 11, 5099-5112, 10.5194/acp-11-5099-2011, 2011.

Nguyen, L. S. P., Huang, H.-Y., Lei, T. L., Bui, T. T., Wang, S.-H., Chi, K. H., Sheu, G.-R., Lee, C.-T., Ou-Yang, C.-F., and Lin, N.-H.: Characterizing a landmark biomass-burning event and its implication for aging processes during long-range transport, *Atmospheric Environment*, 241, 117766, https://doi.org/10.1016/j.atmosenv.2020.117766, 2020.

Ogren, J. A., Wendell, J., Andrews, E., and Sheridan, P. J.: Continuous light absorption photometer for long-term studies, *Atmos. Meas. Tech.*, 10, 4805-4818, 10.5194/amt-10-4805-2017, 2017.

Ohata, S., Schwarz, J. P., Moteki, N., Koike, M., Takami, A., and Kondo, Y.: Hygroscopicity of materials internally mixed with black carbon measured in Tokyo, *Journal of Geophysical Research: Atmospheres*, 121, 362-381, https://doi.org/10.1002/2015JD024153, 2016.

Pöhlker, M. L., Pöhlker, C., Quaas, J., Mülmenstädt, J., Pozzer, A., Andreae, M. O., Artaxo, P., Block, K., Coe, H., Ervens, B., Gallimore, P., Gaston, C. J.,

Gunthe, S. S., Henning, S., Herrmann, H., Krüger, O. O., McFiggans, G., Poulain, L., Raj, S. S., Reyes-Villegas, E., Royer, H. M., Walter, D., Wang, Y., and Pöschl, U.: Global organic and inorganic aerosol hygroscopicity and its effect on radiative forcing, *Nature Communications*, 14, 6139, 10.1038/s41467-023-41695-8, 2023.

Quaas, J., Arola, A., Cairns, B., Christensen, M., Deneke, H., Ekman, A. M. L., Feingold, G., Fridlind, A., Gryspeerdt, E., Hasekamp, O., Li, Z., Lipponen, A., Ma, P. L., Mülmenstädt, J., Nenes, A., Penner, J. E., Rosenfeld, D., Schrödner, R., Sinclair, K., Sourdeval, O., Stier, P., Tesche, M., van Diedenhoven, B., and Wendisch, M.: Constraining the Twomey effect from satellite observations: issues and perspectives, *Atmos. Chem. Phys.*, 20, 15079-15099, 10.5194/acp-20-15079-2020, 2020.

Schmeisser, L., Andrews, E., Ogren, J. A., Sheridan, P., Jefferson, A., Sharma, S., Kim, J. E., Sherman, J. P., Sorribas, M., Kalapov, I., Arsov, T., Angelov, C., Mayol-Bracero, O. L., Labuschagne, C., Kim, S. W., Hoffer, A., Lin, N. H., Chia, H. P., Bergin, M., Sun, J., Liu, P., and Wu, H.: Classifying aerosol type using in situ surface spectral aerosol optical properties, *Atmos. Chem. Phys.*, 17, 12097-12120, 10.5194/acp-17-12097-2017, 2017.

Wang, S.-H., Lin, N.-H., Chou, M.-D., and Woo, J.-H.: Estimate of radiative forcing of Asian biomass-burning aerosols during the period of TRACE-P, *Journal of Geophysical Research: Atmospheres*, 112, https://doi.org/10.1029/2006JD007564, 2007.

Yen, M.-C., Peng, C.-M., Chen, T.-C., Chen, C.-S., Lin, N.-H., Tzeng, R.-Y., Lee, Y.-A., and Lin, C.-C.: Climate and weather characteristics in association with the active fires in northern Southeast Asia and spring air pollution in Taiwan during 2010 7-SEAS/Dongsha Experiment, *Atmospheric Environment*, 78, 35-50, https://doi.org/10.1016/j.atmosenv.2012.11.015, 2013.

---

## Author Comment (AC2)

**We appreciate the suggestion from the referee. Reviewer reports are marked in black font, our responses are marked in green bold font, and the changes to the revised manuscript are marked in blue bold font.**

- The title is a bit hard to read. I'd suggest revising it to "Separating the Twomey effect and the semi-direct effect through the cloud–aerosol mixing ratio," which more clearly conveys the main points of the paper.

**We appreciate the reviewer's constructive suggestion. The original title was indeed not sufficiently concise. However, distinguishing the aerosol-cloud interactions specifically under absorbing-aerosol conditions is a central theme of this study. Although absorbing aerosols are implicitly related to the semi-direct effect, we believe that explicitly stating this context is important to avoid misunderstanding the cloud–aerosol mixing ratio as a metric for differentiating absorbing versus non-absorbing aerosols. Clarifying that both the Twomey effect and the semi-direct effect are examined within absorbing-aerosol environments therefore strengthens the accuracy of the title. If title modification is permitted, we are willing to revise it to:**
**"Separating the Twomey Effect and the Semi-Direct Effect in Absorbing Aerosol Environments through the Cloud-Aerosol Mixing Ratio."**

- Abstract:
1  The term "nonlinear" should be clearly defined, especially since it is emphasized in the title. After reading the abstract, it remains unclear whether it refers to the nonlinear response to aerosol amount or to MCr.
2  Including simple equations for the key metrics (the cloud–aerosol ratio and the ACI index) would be helpful. In particular, the interpretation of the ACI index depends on whether it is defined in terms of droplet number concentration (Nd) or droplet size (https://doi.org/10.1029/2008JD011006).
3  I think it's nice to also mention ACI index under low MCr.
- L16: better ->

**Below are the revisions we made to the abstract.**

**L15-20:   …This study applies the cloud-aerosol mixing ratio (e.g., mass concentration mixing ratio (MCr), "$log\left(\frac{LWC \times 10^6}{PM_{2.5}}\right)$") in conjunction with the ACI**

index "($ACI_{N_d} = \frac{1}{3}\left(\frac{\partial \ln N_d}{\partial \ln \alpha}\right)$)" to describe the behavior of aerosol-cloud interactions. Results identify … (2) under low MCr conditions (MCr = 3-4.5), Not only is the $ACI_{N_d}$ less than -0.06, but the high aerosol loading … This study provides a comprehensive explanation of how "absorbing aerosols" influence cloud systems over East Asia and highlights "the critical role of the cloud-aerosol mixing ratio in characterizing the microphysical responses associated with the Twomey effect and the semi-direct effect."

- L45-50: It would be good to note that such nonlinear behavior has been widely observed from space recently, and understanding it is crucial for improving climate predictions, especially in the context of continuously decreasing aerosol emissions (https://doi.org/10.1038/s41558-023-01775-5).

We have added this study to the reference list and moved L80-81 to follow the discussion of that paper, thereby improving the logical flow of the manuscript.

L49: …effect on cloud droplet size and number "(Feingold et al., 2001; Saponaro et al., 2017; Chen et al., 2021; Jia and Quaas, 2023)". "Observational studies have also shown that the water content in the atmosphere plays a critical role in determining whether aerosols ultimately enhance or suppress cloud formation (Zhang and Zuidema, 2019, 2021). Furthermore," some cloud…

L80-81:

- L62-75: This section discusses why the quantification of ACI remains so uncertain. I think it would benefit from referencing some more timely review papers that summarize the main sources of uncertainty in the Twomey effect (e.g., https://doi.org/10.5194/acp-20-15079-2020, https://doi.org/10.5194/acp-22-7353-2022).

We have rewritten the relevant section and incorporated a discussion on the uncertainties associated with the Twomey effect.

L62: To quantify the impact of aerosols on cloud properties, previous studies have utilized aerosol-cloud interaction (ACI) index as a measure (Kaufman and

Fraser, 1997; Feingold et al., 2001; McComiskey et al., 2009; Lihavainen et al., 2010; Zheng et al., 2020; Chen et al., 2021). "In particular, positive ACI index values (ranging from 0 to 0.33) are generally interpreted as an indication of the strength of the Twomey effect (McComiskey et al., 2009; Lihavainen et al., 2010)." However, … to specific liquid water regimes (McComiskey et al., 2009).

"In addition, studies based primarily on satellite data are often constrained by retrieval limitations, as simultaneous estimation of cloud and aerosol properties within the same spatial domain remains challenging (Jia et al., 2021). Moreover, variations in cloud-base and cloud-top CCN concentrations, uncertainties in $N_d$, and the low vertical resolution of aerosol optical depth (AOD) introduce substantial uncertainty in estimating in-cloud aerosol concentrations (Quaas et al., 2020; Jia et al., 2022). Furthermore, variations in aerosol composition, number concentration, hygroscopicity, and optical properties (Wang et al., 2010; McMeeking et al., 2011; Ohata et al., 2016; Pöhlker et al., 2023) make the conversion from aerosols to CCN difficult to quantify (Quaas et al., 2020), significantly increasing the complexity of retrieving and interpreting aerosol-cloud interaction signals. These factors contribute to the persistent difficulty in robustly quantifying aerosol effects on clouds. Consequently, observational ACI estimates frequently demonstrate significant spatial and temporal constraints, and may not generalise effectively across varied cloud regimes or environmental conditions (McComiskey et al., 2009; Lihavainen et al., 2010; Quaas et al., 2020; Jia et al., 2022)." To present, …

- L92: a definition of cloud water-aerosol mixing ratio should be given here'

L92: Furthermore, "this study introduces the use of the cloud-aerosol mixing ratio to represent how much liquid water is present per unit of aerosol concentration, thereby providing a more realistic indication of aerosol loading within cloud systems." This framework allows for a systematic examination of the transition between Twomey effect-dominated …

- L184: α refers to aerosol parameters such as AOD, PM2.5, or aerosol number concentration (Na) à α refers to proxies for CCN number concentration, such as AOD, aerosol index, sulphate mass concentration, PM2.5, or aerosol number concentration (Na)… many widely used CCN proxies should be metioned.
- L194: cloud number concentration -> cloud droplet number concentration
- L195: is less constrained by LWC -> does not rely on the fixed-LWC assumption

We have integrated this discussion into L194 to improve the logical flow of the manuscript, and we have incorporated the CCN proxies suggested by the reviewer.

L184: Here, Nd is the cloud droplet number concentration, COT is the cloud optical thickness, and …

L194: "Here, $N_d$ is the cloud droplet number concentration, COT is the cloud optical thickness, and LWP is the liquid water path, while α refers to proxies for CCN number concentration, such as AOD, aerosol index (AI), sulphate mass concentration, PM2.5, or aerosol number concentration ($N_a$)." In contrast, McComiskey et al. (2009) indicated that the "cloud droplet number concentration" emphasizes the activation process from aerosol to cloud droplet. This metric, particularly when direct measurements of droplet number are available, "does not rely on the fixed-LWC assumption". While Eq. (3) …

- Section 2.4: The Nd-to-aerosol susceptibility has become more commonly used than the ACI index in recent years (e.g., https://doi.org/10.1038/s41467-018-05028-4, https://doi.org/10.5194/acp-20-15079-2020). I wouldn't suggest removing this part, but perhaps linking ACINd to the Nd-to-aerosol susceptibility would be a good compromise.

Thank you for the reviewer's insightful suggestion. We present the reorganized discussion below.

L199: …into cloud droplets. "In recent years, the calculation formula derived from the ACI index has been increasingly regarded as one of the approaches to quantify the sensitivity of cloud systems to aerosol perturbations (Ma et al., 2018; Liu et al., 2024). In particular, the $N_d$-to-aerosol susceptibility provides a straightforward representation of the relationship between aerosol variations and changes in cloud droplet number (Quaas et al., 2020), thereby giving the index a clear physical meaning."

- Figure 3(b): what does the 'differential results' mean? Is it the derivative of curve in fig3a? Would be good to have an equation for this.

This is indeed the derivative of the curve shown in Figure 3(a). Our intention is to more clearly illustrate the differences that arise from selecting different time

**intervals for the calculation. To achieve this, we computed the values in a discrete manner, and the corresponding equation has been added in the updated Figure 3 (b).**

[Figure]

**Figure 3**: (a) the standard error of the mean for PM$_{2.5}$ (black), LWC (orange), and N$_d$ (blue) at different time resolutions; (b) the differential results of the standard error of the mean.

- Section 2.6: SEM sounds a smart idea to remove the high-frequency signal. Could the authors explain a bit more about its physical meaning? And, the use of 60 minute is nicely justified, but I wonder to what extent the time interval can change the ACI results in section 3.

**When the selected time interval is too short, the calculations may be dominated by excessive fluctuations, and the number of data points becomes insufficient to ensure that the derived ACI index is statistically meaningful. Conversely, if the interval is too long, variations in meteorological conditions may overshadow the aerosol-related signals. Therefore, when examining the temporal evolution of aerosol–cloud interactions, our core objective is to identify the shortest reasonable time interval that preserves the physical linkage between aerosols and clouds while ensuring statistical robustness.**

The results in Figure A1 clearly demonstrate this behavior. When the averaging interval exceeds 120 minutes (Fig. A1(d)–(f)), most ACI index values cluster around zero, indicating a loss of sensitivity to aerosol–cloud interaction signals. In addition, for some cloud events (Events 3 and 4), a single long averaging window can span nearly the entire event. In such cases, the resulting distributions may reflect differences in meteorological conditions between cloud events rather than aerosol-induced perturbations within an event.

[Figure]

Figure A1: Scatter and density plots of the MCr-ACI index for the continuous

cloud events. The density plot illustrates the overall distribution of all calculated results, while the scatter plot shows the distribution of individual cloud events using different colors and symbols (one point shown for every 30 calculations). (a)-(f) represent different sizes of the sampling time intervals.

- Figure 4 & Table2: I suggest to plot the duration of each event (shown in Table 2) in Fig. 4, making it easier to read L279-284

We thank the reviewer for the suggestion. We have reorganized the discussion and included the corresponding meteorological information at the time of observation in the supplementary (Fig. S2) material to aid readability. In addition, we have rewritten the section to frame the analysis around the evolution of cloud events, allowing a more focused comparison of whether aerosol optical properties are consistent across different cloud events.

In Figure S2(c), it can be seen that not all periods with low visibility and relative humidity near 100% were classified as continuous cloud events. This is because some of these periods were caused solely by precipitation, and sometimes the CDP laser windows were wet or contaminated by rain, resulting in poor-quality observations that were excluded prior to analysis.

**L261:**

Figure 4 presents the aerosol observations from the LABS aerosol system, with the corresponding meteorological information shown in Fig. S2. In this section, the aerosol optical properties of aerosols are analyzed using PM10 measurements from the aerosol system, whereas PM2.5 observations from the 11-D instrument are primarily used in the other sections. The gray-shaded intervals denote continuous cloud events lasting more than three hours and detected by CDP (Table 2). Based on the absorption and scattering coefficients as well as aerosol mass concentrations (Fig. 4 (a), (b) and Fig. S2 (d)), it is evident that during cloud events accompanied by significant precipitation (Fig. S2 (d)), aerosol concentrations decreased due to washout effect. However, in periods without precipitation signals prior to 10 March, aerosol concentrations clearly increased, indicating continuous long-range transport of aerosols to the LABS region. Overall, higher pollutant levels were observed during the early stage of the cloud event on March 1 (Event 1), both the early and late stages of the event on March 7 (Event 2), and throughout the event on March 9 (Event 3). In contrast, lower concentrations occurred during the precipitation periods on March 1 and March 7, and during the cloud events on March 10 and 11 (Event 4

and Event 5).

Further analysis was conducted using AAE and SAE to evaluate the light absorption and scattering abilities of aerosols at blue and red wavelengths (Fig. 4 (c), (d)). Except for the Event 1, when the mean AAE value was significantly lower than 1 (0.61 ± 0.12), the average AAE during other cloud events was close to 1, suggesting minimal wavelength dependence of aerosol absorption. The SAE remained relatively stable throughout the observation period but showed a slight decrease on March 9 (1.47 ± 0.01), possibly indicating the presence of coarse-mode aerosols related to local sources. The single-scattering albedo (SSA), used to evaluate the overall aerosol absorption capacity (Fig. 4(e)), was generally below 0.9 during most of the observation period except on March 12. Notably, no significant correlation was found between SSA values and aerosol concentrations before and after cloud events, suggesting that despite effects of wet deposition, the proportion of light-absorbing aerosols within the total aerosol population remained relatively consistent. In some cloud events (Events 2 and 5), SSA even decreased during precipitation (Fig. S2 (d)), indicating an increased fraction of absorbing aerosols, likely due to the relatively hydrophobic nature of black carbon (McMeeking et al., 2011; Ohata et al., 2016; Pöhlker et al., 2023), which reduces its susceptibility to wet removal compared to more hygroscopic components.

The overall…

Table 2: the start, end local times and the mean of $PM_{2.5}$, LWC of continuous cloud events (UTC +8).

| Cloud Event | Start Time (LT) | End Time (LT) | Average $PM_{2.5}$ | Average LWC |
|---|---|---|---|---|
| Event 1 | 3/1 09:37 | 3/1 15:40 | 7.8 ± 3.1 | 0.130 ± 0.130 |
| Event 2 | 3/7 09:16 | 3/8 07:28 | 11.4 ± 9.3 | 0.138 ± 0.126 |
| Event 3 | 3/9 11:07 | 3/9 15:27 | 20.3 ± 4.5 | 0.163 ± 0.164 |
| Event 4 | 3/10 02:13 | 3/10 06:00 | 2.3 ± 1.6 | 0.229 ± 0.085 |
| Event 5 | 3/11 09:47 | 3/12 05:21 | 2.5 ± 3.2 | 0.147 ± 0.121 |

[Figure]

**Figure 4: PM₁₀ aerosol optical properties observed by the aerosol system at LABS, the gray-shaded intervals represent continuous cloud events. (a) absorption coefficient; (b) scattering coefficient; (c) Absorption Ångström Exponent (AAE, red) and Scattering Ångström Exponent (SAE, orange); (d) Single Scattering Albedo (SSA), and gray dash line means SSA = 0.9.**

[Figure]

**Figure S2: Hourly averages of meteorological parameters and aerosol mass concentrations observed at LABS from March 1 to 12, 2024: (a) wind speed and direction; (b) temperature (green) and pressure (red); (c) visibility (orange) and relative humidity (dark blue); (d) PM$_{10}$ (dark red) and PM$_{2.5}$ (red) measured by the 11-D instrument, along with hourly precipitation (light blue bars).**

- L287-289: 'Due to significant…between high and low aerosol loads.' I don't quite see how this statement is relevant here. As I understand it, all aerosol and cloud measurements were made at the same ground level at this mountain site, so vertical co-location shouldn't be an issue in this study.

**At LABS, a high-altitude background station, the baseline concentrations of pollutants are substantially lower than those typically observed in urban or suburban environments influenced by anthropogenic emissions, provided that no**

significant long-range transport or strong vertical transport mechanisms (e.g., subsidence or valley winds) occur. Moreover, the diurnal variability of pollutants at LABS differs markedly from that within the boundary layer. Therefore, the conventional definition of a "pollution event" is not directly applicable to this site. We agree with the reviewer that the original statement may have been difficult to interpret, and the revised version is presented below for clarity.

L288: … served as the distinguishing criterion. "Because LABS is situated at a high altitude within the free atmosphere, the variability in pollutant concentrations is markedly different from that in the boundary layer dominated by anthropogenic emissions. As a result, conventional air quality indices may not adequately capture the magnitude of aerosol loading observed at the site." Therefore, …

- Figure 6: it's interesting to see the different ACI behaviors in low and high aerosol groups. I wonder if there is significant difference in SSA between two groups, which will help to understand the shift.

Following the reviewer's suggestion, we analyzed the minute-resolution data from the aerosol system at LABS and plotted the results in Figure A2. As shown in Figures A2(a) and (b), the mean differences in both AAE and SAE across different aerosol concentration ranges are minimal. Most data points can be categorized as BC-type aerosols based on the optical characteristics (Schmeisser et al., 2017). It is worth noting that the standard deviation of AAE is larger in the low-concentration range ($PM_{2.5} < 11.1$ μg m$^{-3}$), which can be partly attributed to the increased uncertainty of the CLAP instrument under low absorption coefficients and high temporal resolution conditions (Ogren et al., 2017). For the comparison of SSA (Fig. A1(c)), the SSA values remain consistently below 0.9 regardless of aerosol concentration, indicating that the proportion of absorbing aerosols in the environment is relatively stable. In particular, the fraction of absorbing aerosols does not increase under low-concentration conditions, confirming that the consistency of aerosol optical properties does not affect the validity of the analysis in this study. Since the aerosol system at LABS is designed for long-term monitoring, we prefer to use hourly averaged data for better QA/QC reliability. Additionally, the system alternately measures $PM_{10}$ and $PM_1$ every 30 minutes, meaning that only up to 30 minutes of data are available for each size fraction per hour. This characteristic is also reflected in the sample counts of each group shown in Figure A2.

According to the reviewer's recommendation, we further analyzed the aerosol characteristics inside and outside of cloud events in Sect. 3.1 of the revised manuscript (see precious responses (Fig. 4)). The conclusions are consistent with the results in Figure A2. To keep the subsequent analysis more focused on the variations in cloud microphysical properties, we prefer to retain the original set of variables in Figure 6 of the manuscript.

For clarity, we have revised Figure 6 into six subpanels and modified the x-axis to display $PM_{2.5}$ concentrations. This adjustment more clearly highlights the concentration differences among the groups and improves overall readability.

[Figure]

**Figure A2: Aerosol optical properties categorized into 10 groups of equal sample size based on $PM_{2.5}$ obtained from 11-D (the data number in each group is shown below as gray bars and values). (a) AAE, (b) SAE, (c) SSA.**

[Figure]

**Figure 6: Cloud microphysical and weather data categorized into 10 groups of equal sample size based on PM$_{2.5}$ obtained from 11-D (the data number in each group is shown below as gray bars and values). (a) N$_d$, (b) ED, (c) LWC from CDP observation results; (d) temperature, (e) relative humidity and (f) specific humidity are weather results measured at LABS.**

- L332: why 4.5 not 4.64 - the minimal MCr with ACI index>0?

From our perspective, 4.64 indeed represents the minimum value derived from the analysis. However, from the standpoint of real-world observations, we believe that additional measurements, across different regions, meteorological conditions, and aerosol types, are necessary before the MCr threshold can be more firmly constrained. Defining the value too rigidly at this stage would not strengthen the main message of this study. Moreover, variations in instruments, sampling strategies, and site characteristics across global observatories may

introduce uncertainties, and an overly precise value (with many decimal places) would likely reflect these uncertainties rather than provide meaningful physical insight, thereby distracting from the key findings.

Nevertheless, to demonstrate the robustness of using 4.5 as a representative threshold in our study, we further evaluate and validate this choice through analyzes conducted across different MCr ranges in Sect. 3.3.

- L332-333: This sentence is a bit confusing. could you clarify from what to what the proportion increases from 13% to 30%?
- L337-338: A very confusing part again… 'Although the ACI shifted from negative to positive as aerosol concentrations increased': however, from the figure, it seems that the negative-to-positive shift occurs as MCr increases, not aerosol concentration. Also, when MCr < 2.5, the proportion of aerosol should be lower than that of cloud water, shouldn't it? Please clarify these.

We agree that the descriptions in these two parts were not sufficiently precise in the original manuscript.

1. The comparison of 13% and 30% refers to the fraction of cases in which the ACI index is positive. Specifically, 13% represents the proportion of all calculated ACI values that are positive, whereas 30% represents the proportion of positive ACI values when only cases with $MCr > 4.5$ are considered. This distinction highlights that positive $ACI_{Nd}$ are predominantly concentrated in the higher MCr regime.

2. In this study, MCr inherently accounts for both aerosol loading and liquid water content within the cloud system. Physically, it represents how much liquid water corresponds to each unit of aerosol concentration. Therefore, MCr is not simply a direct comparison between $PM_{2.5}$ and LWC, as the earlier phrasing may have implied.

To improve clarity, we revised and streamlined the discussion in Section 3.3 and reorganized the structure. We also added Table 2, which provides the mean and standard deviation of $PM_{2.5}$ and LWC for different cloud events. In the latter part of Section 3.2, we now focus on how variations in $PM_{2.5}$, LWC, and MCr across cloud events correspond to changes in the ACI values, thereby emphasizing the central role of MCr. For a more focused presentation, Figure 7 now shows only the MCr-based results, while the NCr results have been moved to the supplementary material (Fig. S3).

**L319-331:**

L323: … display a positive ACI index value. "Of these, over 95% (334 out of 351) occurred in regions where MCr exceeded 4.5 (with the minimum MCr value within the reasonable range for ACI index > 0 being 4.64), especially when positive ACI values within the reasonable range (0-0.33, Fig. 7). This finding suggests that under the influence of high aerosol concentrations, cloud systems tend to adjust by reducing the cloud droplet number concentrations most of the time. As shown in Figure S3, the distribution of NCr-ACI reveals a similar tendency to that observed in the MCr-ACI relationship. After constraining aerosol properties and meteorological conditions, negative ACI index can be meaningfully interpreted (as described in Sect. 2.4). However, most of the previous studies have focused only on positive region. It is important to note that negative values do not have a defined "reasonable range", they simply indicate that an increase in aerosol concentration is accompanied by a decrease in cloud droplets number." The physical mechanism …

L332-354:
Comparative analysis among individual cloud events reveals several notable trends (Fig. 7). All events show a consistent relationship between MCr and ACI, where smaller ACI values (more negative) correspond to lower liquid water content per unit aerosol abundance (i.e., smaller MCr). A closer examination of cases with MCr > 4.5 shows that 31% of ACI values (135 data points) are positive in Event 2, and 93% (148 data points) are positive in Event 4. In contrast, although ACI values in Events 1 and 5 tend to increase with MCr, most of them remain negative. These findings suggest that short-term variations in cloud systems are more sensitive to environmental changes induced by absorbing aerosols, where increased aerosol loading can lead to the dissipation of cloud droplets. It is noteworthy that although Event 3 exhibited relatively high mean LWC (the second highest among the five consecutive cloud events), it occurred under high aerosol loading conditions (Table 2). Consequently, the liquid water content per unit aerosol abundance was relatively low (MCr ≈ 4; Fig. 7), and the calculated ACI values remained negative throughout the event (Fig. 7). This result suggests that aerosol-cloud interactions under the influence of absorbing aerosols cannot be interpreted solely based on aerosol concentration. In particular, since the liquid water content varies substantially among real world cloud events, MCr provides a more representative measure of the actual aerosol loading within clouds.

[Figure]

**Figure 7: Scatter and density plots of the MCr-ACI index for the continuous cloud events. The density plot illustrates the overall distribution of all calculated results, while the scatter plot shows the distribution of individual cloud events using different colors and symbols (one point shown for every 30 calculations).**

[Figure]

**Figure S3: Scatter and density plots of the NCr-ACI index for the continuous cloud events. The settings are identical to those used in Figure 7.**

- Figure 7: Including SSA could help explain the role of absorption in evaporation. It might be interesting to make a plot similar to Fig. 7, but with each point colored by its corresponding SSA value.

**The single-scattering albedo (SSA) represents the average absorptivity of the aerosol population and therefore provides a good indication of the fraction of purely absorbing aerosols within the total aerosol mixture. However, for evaluating the environmental relevance of the semi-direct effect, we consider that the absolute amount of absorbing aerosols is more important than their relative proportion.**

**Figure A3 presents the absorption coefficient derived from the CLAP $PM_{10}$ measurements (Fig. A3 (a)) together with the minute-resolution SSA retrieved from the aerosol system (Fig. A3 (b)). Although SSA tends to be lower in the MCr > 4.5 regime (discussion in Fig. A2) the corresponding absorption coefficients remain very small. This is because aerosol concentrations in this regime are intrinsically low during our field campaign. In contrast, for MCr < 4.5, the absorption coefficient is noticeably higher despite the larger SSA values.**

**Nevertheless, the interpretation of the ACI index cannot rely solely on aerosol loading. The relevant discussion, including how these differences reflect aerosol-cloud interactions under different mixing ratios, has been clarified in the revised Section 3.3.**

[Figure]

**Figure A3: Scatter plots of the MCr-ACI index for the continuous cloud events. The color scale represents (a) the mean CLAP absorption coefficient at 528 nm and (b) the mean single-scattering albedo (SSA).**

- L32: 'cloud optical thickness' -> cloud optical thickness and thus cloud albedo

L32: **…a greater number of cloud droplets with smaller droplet sizes will form, leading to an increase in cloud optical thickness "and thus cloud albedo", which …**

- L40: comparing to cloud amount, LWP is even more uncertain and should be mentioned here

L40: **…instead enhance evaporation (Chen et al., 2012; Fan et al., 2016; Toll et al., 2019). "This not only indicates large uncertainties in the estimation of LWP but may also lead to misinterpretations in models regarding aerosol-induced variations in LWP and cloud fraction (CF) (Toll et al., 2019; Chen et al., 2022)." It is worth noting that…**
- L57: suppress supersaturation - > reduce ambient supersaturation

L57: **…absorbing aerosols may "reduce ambient supersaturation levels" …**

- L233-234: Not to mention that RH measurements cannot capture supersaturation conditions: I didn't get this sentence.

**The purpose of this statement is to clarify the measurement limitations of the RH sensor. Specifically, the RH instrument cannot detect supersaturated conditions (RH > 100%), and thus the variations that occur beyond saturation cannot be captured by the sensor. Below are the revisions of the sentence.**

L233: **"Furthermore, conventional relative humidity instruments cannot capture variations under supersaturated conditions (RH > 100%)." While LWC also does not directly quantify supersaturation …**

**In addition, we refined the textual descriptions in Sections 3.2, 3.4 and 3.5 to improve clarity and readability. In Section 3.2, the descriptions of the groups were revised to directly reference $PM_{2.5}$ concentration, making the interpretation more intuitive. In Section 3.4 and 3.5, we reorganized and streamlined theoretical discussions that had been mentioned multiple times to enhance the logical flow and avoid redundancy.**

**Sect. 3.2:**

**L292: Figure 6 shows the aerosol loading in the environment, ranging from low to high. "The maximum mean value of $N_d$ (370 cm$^{-3}$) was observed when the PM$_{2.5}$ concentration ranged between 1.4 and 2.6 µg m$^{-3}$ (Fig. 6 (a)), whereas LWC reached the maximum mean value (0.219 g m$^{-3}$) within the lower PM2$_{2.5}$ concentration range of 0.4-1.4 µg m$^{-3}$(Fig. 6 (c)). Notably, when PM$_{2.5}$ ≥ 2.6 µg m$^{-3}$, both $N_d$ and LWC exhibited a decreasing trend as environmental aerosol concentrations increased. The variation in effective diameter (ED) can be characterized by a threshold at PM$_{2.5}$ = 5.7 µg m$^{-3}$ (Fig. 6 (b)). When PM$_{2.5}$ concentrations were below this value, ED decreased with increasing aerosol loading. However, when PM$_{2.5}$ ≥ 5.7 µg m$^{-3}$, ED maintained an average size or slightly increased despite the higher aerosol concentration. A further comparison of weather data across different PM$_{2.5}$ concentration ranges revealed that when PM$_{2.5}$ ≥ 21.7 µg m$^{-3}$, the average temperature (4.3°C, Fig. 6 (d)) and specific humidity (7.0 g kg$^{-1}$, Fig. 6 (f)) reached the lowest values, possibly constrained by specific time and weather conditions. Additionally, relative humidity (Fig. 6 (e)) displayed a decreasing trend with increasing environmental aerosol loading, and the standard deviation of relative humidity increased, reaching the lowest value (93.4%) and the highest standard deviation (7.1%) between PM$_{2.5}$ concentrations of 16.4 and 21.7 µg m$^{-3}$."**

**L305: … below 4.2 µg m$^{-3}$. "Compared with environments where PM$_{2.5}$ ≥ 5.7 µg m$^{-3}$, those with extremely low aerosol concentrations (PM$_{2.5}$ < 2.6 µg m$^{-3}$) exhibited noticeably higher LWC, highlighting the substantial differences in cloud liquid water among different aerosol loading conditions. The range of PM$_{2.5}$ between 2.6 and 5.7 µg m$^{-3}$ appeared to represent a transitional phase. In environments with relatively low average aerosol concentrations, higher LWC coexisted with higher $N_d$ and lower ED values, indicating that sufficient water availability allowed more aerosol particles to be activated." It is important to note that … In the high aerosol loading environment "(PM$_{2.5}$ ≥ 5.7 µg m$^{-3}$)," increased aerosol concentrations result in a decrease in $N_d$ and a slight increase in ED …**

**L460: "In a comparative analysis involving 10 data groups stratified by both PM$_{2.5}$ concentration levels and sample size considerations, a distinct trend emerged: in the low-concentration (PM$_{2.5}$ < 2.6 µg m$^{-3}$), an increase in PM$_{2.5}$ was associated with an increase in $N_d$ and a significant decrease in ED. However, in the high-concentration when PM$_{2.5}$ ≥ 5.7 µg m$^{-3}$, further increases in PM$_{2.5}$ corresponded to a decrease in $N_d$ and a slight increase in ED. These results**

suggest a transition in cloud microphysical responses under elevated aerosol loading conditions." …

**Sect. 3.4:**

L389: ~~As observed in Fig. 6, an increase in temperature is also noted in Groups 6-9. The reduction in environmental supersaturation primarily affects smaller cloud droplets, which may fail to sustain growth or even be unable to surpass the Köhler critical point before developing into clouds. Therefore, as aerosol loading transitions from low to high, the microphysical effects of increasing absorbing aerosols on clouds shift from the Twomey effect to semi-direct effect.~~

**Sect. 3.5:**

L397: …MCr grouping. "For each group, data were further separated into high and low aerosol loading based on the median PM2.5 concentration. In MCr [4.5, 6], high aerosol loading yields a markedly higher frequency of $N_d$ between 300-600 cm$^{-3}$ compared with both the overall observations and the low-aerosol subset (Fig. 10 (c)). In the frequency distribution of ED (Fig. 10 (b)), it was also found that frequencies below 12 μm are significantly higher, which is consistent with the results in Sect. 3.4." In the MCr [3, 4.5] range …

L409: …across these regimes. "However, restricting the LWC range is necessary to reduce uncertainties when assessing aerosol effects on droplet size." Considering weather conditions, Table 3 shows the mean value of temperature, relative humidity, specific humidity in different constraint conditions. "In the high aerosol loading of MCr [3, 4.5]," the temperature…

L420: …20 μm are observed under high aerosol load. "When LWC is restricted (Fig. 12 (a)), the aerosol size distribution remains similar except for fewer particles larger than 2.5 μm." Under high aerosol loading, an increase in cloud droplets smaller than 12 μm is observed, while the larger droplets (> 12 μm) markedly decrease.  "Constraint LWC (Fig.12 (a)) more clearly characterizes the Twomey effect, illustrating how aerosol particle concentrations influence the droplet size distribution, "in particular, there are minimal differences between high and low aerosol pollution in terms of weather

conditions (Table 3)."

L439: …~~dividing point. Combined with the observation of larger coarse-mode aerosol particles in high aerosol loading conditions, this suggests that coarse-mode aerosols can activate and grow into larger droplets under lower supersaturation conditions, but small droplets are more prone to dissipation, consistent with their position on the Köhler curve (Klemm and Lin, 2016). In terms of weather conditions, under constrained LWC, the high aerosol pollution interval in MCr [3, 4.5] has a mean temperature of 6.8°C, relative humidity of 94%, and specific humidity of 8.02 g kg⁻¹. Under low aerosol pollution, the mean temperature is 5.8°C, relative humidity is 98%, and specific humidity is 7.80 g kg⁻¹. Differences in these results agree with the previous discussion (Table 3) in this section.~~

L448: …the interpretation becomes more complex. "Cloud droplet size distributions can be explained by    Köhler curve, coarse-mode aerosols under high aerosol loading can more easily activate and grow into larger droplets under lower supersaturation conditions, while smaller droplets are more prone to evaporation (Klemm and Lin, 2016)." Although …

**Reference**

Chen, Y., Haywood, J., Wang, Y., Malavelle, F., Jordan, G., Partridge, D., Fieldsend, J., De Leeuw, J., Schmidt, A., Cho, N., Oreopoulos, L., Platnick, S., Grosvenor, D., Field, P., and Lohmann, U.: Machine learning reveals climate forcing from aerosols is dominated by increased cloud cover, *Nature Geoscience*, 15, 609-614, 10.1038/s41561-022-00991-6, 2022.

Chen, Y. C., Christensen, M. W., Xue, L., Sorooshian, A., Stephens, G. L., Rasmussen, R. M., and Seinfeld, J. H.: Occurrence of lower cloud albedo in ship tracks, *Atmos. Chem. Phys.*, 12, 8223-8235, 10.5194/acp-12-8223-2012, 2012.

Chen, Y. C., Wang, S. H., Min, Q., Lu, S., Lin, P. L., Lin, N. H., Chung, K. S., and Joseph, E.: Aerosol impacts on warm-cloud microphysics and drizzle in a moderately polluted environment, *Atmos. Chem. Phys.*, 21, 4487-4502, 10.5194/acp-21-4487-2021, 2021.

Fan, J., Wang, Y., Rosenfeld, D., and Liu, X.: Review of Aerosol–Cloud Interactions: Mechanisms, Significance, and Challenges, *Journal of the Atmospheric Sciences*, 73, 4221-4252, https://doi.org/10.1175/JAS-D-16-

0037.1, 2016.

Feingold, G., Remer, L. A., Ramaprasad, J., and Kaufman, Y. J.: Analysis of smoke impact on clouds in Brazilian biomass burning regions: An extension of Twomey's approach, *Journal of Geophysical Research: Atmospheres*, 106, 22907-22922, https://doi.org/10.1029/2001JD000732, 2001.

Jia, H. and Quaas, J.: Nonlinearity of the cloud response postpones climate penalty of mitigating air pollution in polluted regions, *Nature Climate Change*, 13, 943-950, 10.1038/s41558-023-01775-5, 2023.

Jia, H., Ma, X., Yu, F., and Quaas, J.: Significant underestimation of radiative forcing by aerosol–cloud interactions derived from satellite-based methods, *Nature Communications*, 12, 3649, 10.1038/s41467-021-23888-1, 2021.

Jia, H., Quaas, J., Gryspeerdt, E., Böhm, C., and Sourdeval, O.: Addressing the difficulties in quantifying droplet number response to aerosol from satellite observations, *Atmos. Chem. Phys.*, 22, 7353-7372, 10.5194/acp-22-7353-2022, 2022.

Kaufman, Y. J. and Fraser, R. S.: The Effect of Smoke Particles on Clouds and Climate Forcing, *Science*, 277, 1636-1639, doi:10.1126/science.277.5332.1636, 1997.

Klemm, O. and Lin, N. H.: What Causes Observed Fog Trends: Air Quality or Climate Change?, *Aerosol and Air Quality Research*, 16, 1131-1142, 10.4209/aaqr.2015.05.0353, 2016.

Lihavainen, H., Kerminen, V. M., and Remer, L. A.: Aerosol-cloud interaction determined by both in situ and satellite data over a northern high-latitude site, *Atmos. Chem. Phys.*, 10, 10987-10995, 10.5194/acp-10-10987-2010, 2010.

Liu, J., Zhu, Y., Wang, M., Rosenfeld, D., Cao, Y., and Yuan, T.: Cloud Susceptibility to Aerosols: Comparing Cloud-Appearance Versus Cloud-Controlling Factors Regimes, *Journal of Geophysical Research: Atmospheres*, 129, e2024JD041216, https://doi.org/10.1029/2024JD041216, 2024.

Ma, P.-L., Rasch, P. J., Chepfer, H., Winker, D. M., and Ghan, S. J.: Observational constraint on cloud susceptibility weakened by aerosol retrieval limitations, *Nature Communications*, 9, 2640, 10.1038/s41467-018-05028-4, 2018.

McComiskey, A., Feingold, G., Frisch, A. S., Turner, D. D., Miller, M. A., Chiu, J. C., Min, Q., and Ogren, J. A.: An assessment of aerosol-cloud interactions

in marine stratus clouds based on surface remote sensing, *Journal of Geophysical Research: Atmospheres*, 114, https://doi.org/10.1029/2008JD011006, 2009.

McMeeking, G. R., Good, N., Petters, M. D., McFiggans, G., and Coe, H.: Influences on the fraction of hydrophobic and hydrophilic black carbon in the atmosphere, *Atmos. Chem. Phys.*, 11, 5099-5112, 10.5194/acp-11-5099-2011, 2011.

Ogren, J. A., Wendell, J., Andrews, E., and Sheridan, P. J.: Continuous light absorption photometer for long-term studies, *Atmos. Meas. Tech.*, 10, 4805-4818, 10.5194/amt-10-4805-2017, 2017.

Ohata, S., Schwarz, J. P., Moteki, N., Koike, M., Takami, A., and Kondo, Y.: Hygroscopicity of materials internally mixed with black carbon measured in Tokyo, *Journal of Geophysical Research: Atmospheres*, 121, 362-381, https://doi.org/10.1002/2015JD024153, 2016.

Pöhlker, M. L., Pöhlker, C., Quaas, J., Mülmenstädt, J., Pozzer, A., Andreae, M. O., Artaxo, P., Block, K., Coe, H., Ervens, B., Gallimore, P., Gaston, C. J., Gunthe, S. S., Henning, S., Herrmann, H., Krüger, O. O., McFiggans, G., Poulain, L., Raj, S. S., Reyes-Villegas, E., Royer, H. M., Walter, D., Wang, Y., and Pöschl, U.: Global organic and inorganic aerosol hygroscopicity and its effect on radiative forcing, *Nature Communications*, 14, 6139, 10.1038/s41467-023-41695-8, 2023.

Quaas, J., Arola, A., Cairns, B., Christensen, M., Deneke, H., Ekman, A. M. L., Feingold, G., Fridlind, A., Gryspeerdt, E., Hasekamp, O., Li, Z., Lipponen, A., Ma, P. L., Mülmenstädt, J., Nenes, A., Penner, J. E., Rosenfeld, D., Schrödner, R., Sinclair, K., Sourdeval, O., Stier, P., Tesche, M., van Diedenhoven, B., and Wendisch, M.: Constraining the Twomey effect from satellite observations: issues and perspectives, *Atmos. Chem. Phys.*, 20, 15079-15099, 10.5194/acp-20-15079-2020, 2020.

Saponaro, G., Kolmonen, P., Sogacheva, L., Rodriguez, E., Virtanen, T., and de Leeuw, G.: Estimates of the aerosol indirect effect over the Baltic Sea region derived from 12 years of MODIS observations, *Atmos. Chem. Phys.*, 17, 3133-3143, 10.5194/acp-17-3133-2017, 2017.

Schmeisser, L., Andrews, E., Ogren, J. A., Sheridan, P., Jefferson, A., Sharma, S., Kim, J. E., Sherman, J. P., Sorribas, M., Kalapov, I., Arsov, T., Angelov, C., Mayol-Bracero, O. L., Labuschagne, C., Kim, S. W., Hoffer, A., Lin, N. H., Chia, H. P., Bergin, M., Sun, J., Liu, P., and Wu, H.: Classifying aerosol type using in situ surface spectral aerosol optical properties, *Atmos. Chem. Phys.*, 17, 12097-12120, 10.5194/acp-17-12097-2017, 2017.

Toll, V., Christensen, M., Quaas, J., and Bellouin, N.: Weak average liquid-cloud-water response to anthropogenic aerosols, *Nature*, 572, 51-55, 10.1038/s41586-019-1423-9, 2019.

Wang, J., Cubison, M. J., Aiken, A. C., Jimenez, J. L., and Collins, D. R.: The importance of aerosol mixing state and size-resolved composition on CCN concentration and the variation of the importance with atmospheric aging of aerosols, *Atmos. Chem. Phys.*, 10, 7267-7283, 10.5194/acp-10-7267-2010, 2010.

Zhang, J. and Zuidema, P.: The diurnal cycle of the smoky marine boundary layer observed during August in the remote southeast Atlantic, *Atmos. Chem. Phys.*, 19, 14493-14516, 10.5194/acp-19-14493-2019, 2019.

Zhang, J. and Zuidema, P.: Sunlight-absorbing aerosol amplifies the seasonal cycle in low-cloud fraction over the southeast Atlantic, *Atmos. Chem. Phys.*, 21, 11179-11199, 10.5194/acp-21-11179-2021, 2021.

Zheng, X., Xi, B., Dong, X., Logan, T., Wang, Y., and Wu, P.: Investigation of aerosol–cloud interactions under different absorptive aerosol regimes using Atmospheric Radiation Measurement (ARM) southern Great Plains (SGP) ground-based measurements, *Atmos. Chem. Phys.*, 20, 3483-3501, 10.5194/acp-20-3483-2020, 2020.